# Geodetic reanalysis of annual glaciological mass balances (2001-2011) of Hintereisferner, Austria

Christoph Klug[1], Erik Bollmann[1], Stephan Peter Galos[2], Lindsey Nicholson[2], Rainer Prinz[3], Lorenzo Rieg[1], Rudolf Sailer[1], Johann Stötter[1], and Georg Kaser[2]

[1]Institute of Geography, University of Innsbruck, Austria
[2]Institute of Atmospheric and Cryospheric Sciences, University of Innsbruck, Austria
[3]Department of Geography and Regional Science, University of Graz, Austria

*Correspondence to:* Christoph Klug (christoph.klug@uibk.ac.at)

**Abstract.** This study presents a reanalysis of the glaciologically obtained annual glacier mass balances at Hintereisferner, Ötztal Alps, Austria for the period 2001-11. The reanalysis is accomplished through a comparison with geodetically derived mass changes, using annual high-resolution airborne laser scanning (ALS). The grid based adjustments for the method-inherent differences are discussed along with associated uncertainties and discrepancies of the two methods of mass balance measure-

ments. A statistical comparison of the two datasets shows no significant difference for seven annual, as well as the cumulative, mass changes over the ten years record. Yet, the statistical view hides significant differences in the mass balance years 2002/03 (glaciological minus geodetic records = +0.92 m w.e.), 2005/06 (+0.60 m w.e.) and 2006/07 (-0.45 m w.e.). We conclude that exceptional meteorological conditions can render the usual glaciological observational network inadequate. Furthermore, we consider that ALS data reliably reproduce the annual mass balance and can be seen as validation or calibration tools for the

glaciological method.

## 1   Introduction

The mass balance of a glacier defines its hydrological reservoir function (e.g., Kaser et al., 2010) and is a reliable indicator of climate change (e.g., Vaughan et al., 2013; Bojinski et al., 2014). Earliest glacier mass balance measurements started around

1950, but only about 30 reference glaciers have uninterrupted annual time series going back to 1976 (e.g., Zemp et al., 2009). This small number of directly measured annual glacier mass balance series provides the basis for reconstructing past contributions to sea level rise (e.g., Kaser et al., 2006; Marzeion et al., 2012; Gardner et al., 2013; Vaughan et al., 2013), extrapolating glacier contribution to regional water supply (e.g., Kaser et al., 2010; Weber et al., 2010; Huss, 2011; Bliss et al., 2014), and glacier change detection, attribution (e.g., Marzeion et al., 2014; Slangen et al., 2017) and projection studies (e.g., Radić and

Hock, 2006; Marzeion et al., 2012; Radić et al., 2014; Huss and Hock, 2015; Mengel et al., 2016).
Since uncertainties and errors in long term mass balance records affect the results of such studies, these must be quantified

and, wherever possible, corrected (e.g., Zemp et al., 2015). Geodetically obtained results have been used as controls for annual glaciological mass balances at decadal scales and are commonly applied to identify random, and to correct systematic, uncertainties in glaciological mass balance time series (Hoinkes, 1970; Haeberli et al., 1998; Fountain and Vecchia, 1999; Krimmel, 1999; Østrem and Haakensen, 1999; Hagg et al., 2004; Cox and March, 2004; Huss et al., 2009; Thibert and Vincent, 2009;

Koblet et al., 2010; Zemp et al., 2010; Prinz et al., 2011; Zemp et al., 2013; Beedle et al., 2014; Galos et al., 2017). Geodetic measurements have also been merged with glaciological mass balance series to increase coverage and representativeness of large regions and global glacier mass balance information (e.g., Cogley, 2009; Gardner et al., 2013). Indeed, the interconnection of different methods is increasingly suggested in order to advance glacier mass change estimates for large regions or even on the global scale (Gardner et al., 2013; Marzeion et al., 2017).

At Hintereisferner in the Austrian Ötztal Alps, glaciological and photogrammetry-based geodetic mass balances are available since the early 1950s (e.g., Kuhn et al., 1999). Early analyses showed good agreement between the two data series on a decadal time scale for the periods 1952/53 to 1963/64 (Lang and Patzelt, 1971) and 1952/53 to 1990/91 (Kuhn et al., 1999). Yet, a more detailed examination by Zemp et al. (2013) revealed discrepancies at Hintereisferner for the periods 1963/64 to 1968/69 and 1978/79 to 1990/91. Between 2001 and 2011, when high resolution air borne laser scanning (ALS) became available, geodetic

mass balances for Hintereisferner were obtained annually. Gross results from the first data pairs indicated considerable differences to the glaciological mass balances (Geist et al., 2007). This motivates a deeper investigation of the apparent discrepancies between the two methods at an annual scale.

Hence, the goal of the present study is to reanalyze the glaciological mass balance record of Hintereisferner for the period 2001 to 2011 and to thereby detect possible shortcomings for individual years or the whole period. We achieve this reanalysis

through a detailed uncertainty assessment using annual geodetic records from high resolution ALS-data. The reanalysis scheme and the assessment of random ($\sigma$) and systematic ($\epsilon$) uncertainties presented in this paper follows the guidelines of Zemp et al. (2013). Hence we refer to this paper for detailed explanations regarding the principal work flow.

## 2   Hintereisferner

Hintereisferner (46.79° N, 10.74° E) is a valley glacier in the Austrian part of the Ötztal Alps (Fig. 1). The glacier consists of three main tributary basins. Langtaufererjochferner (1.11 $km^2$) and Stationsferner (0.28 $km^2$) disconnected from Hintereisferner in 1969 and 2000, respectively, but are still treated as part of the glacier in order to maintain consistency in mass balance assessments over the whole time series of observations. Hence, "Hintereisferner" in this paper refers to all three glacier bodies. The area of Hintereisferner in 2011 was 6.78 $km^2$, about 15% smaller than in 2001, when the first ALS campaign was con-

ducted. The glacier terminus retreated by 390 m during the same period. The glacier elevation ranges from 3720 to 2456 m a.s.l. and the median altitude is 3039 m a.s.l. The accumulation area covers aspects from northeast to southeast while the long and narrow tongue faces northeast. Meltwaters feed the Hintereisbach, which joins the runoff from Kesselwandferner, Hochjochferner and a few smaller glaciers and subsequently drains into Rofenache and finally into the Ötztaler Ache, one of

the major tributaries of the Inn River.

Hintereisferner is located in the "inner dry Alpine zone" (Frei and Schär, 1998), which is among the driest regions of the entire European Alps. Precipitation in Vent (~1900 m a.s.l.), about 8 km west of the glacier terminus, reaches 677 $\mathrm{mm\,a^{-1}}$, with air temperatures of 1.5 °C in average (1906-2011). Precipitation amounts double at the totalizing rain gauge near the Hintereis Research Station (3026 m a.s.l.; Fig. 1), reflecting the altitudinal difference of approximately 1100 m but also the enhanced precipitation activity further up the valley. Over the study period 2001 to 2011, the values for annual temperature and precipitation in Vent are 2.3°C and 676 mm, respectively. The mean annual 0°C-isotherm is located at 2450 m a.s.l.

Like many glaciers in the Eastern European Alps, Hintereisferner has experienced strong shrinkage compared to its Little Ice Age maximum extent, which was reached sometime between 1847 and 1855 (Richter, 1888). Since that time, the glacier area in the Ötztal-Alps has shrunk by more than 50% (Fischer et al., 2015). After a period of rather stationary glacier lengths in the late 1970s and early 1980s (e.g., Patzelt, 1985), glacier mass loss and area shrinkage dominate with particularly high rates during and after the extraordinarily hot summer of 2003 (e.g., Abermann et al., 2009).

## 3  Mass balance methods and data

There are two primary methods for determining the mass balance of a glacier: The glaciological (or direct) and the geodetic method. The glaciological method (e.g., Anonymous, 1969; Hoinkes, 1970; Østrem and Brugman, 1991; Kaser et al., 2003; Cogley et al., 2011) is the most widely used for assessing annual and - more rarely - seasonal mass changes of individual glaciers. It spatially extrapolates in situ point measurements of ablation and accumulation to the glacier-wide surface mass balance, encompassing all mass changes at (near) the glacier surface during the hydrological year (cf. Cogley et al., 2011).

In contrast to the surface mass balance obtained with the glaciological method, the geodetic method differences two consecutive digital elevation models (DEMs) of a glacier and provides its volume change. This method integrates over all processes that lead to surface height changes at any single point of a glacier, i.e. the surface, internal, and basal mass changes as well as those from ice flux divergence, and densification (Cuffey and Paterson, 2010). Consequently, the mass balance values at a certain point of the glacier may differ significantly between the glaciological and the geodetic mass balance method. However, according to the principals of mass conservation, the ice flux divergence becomes zero if integrated over the entire glacier. Moreover, by assuming internal and basal mass changes on mid latitude mountain glaciers to be of minor importance (e.g., Cuffey and Paterson, 2010), and by applying either measured or estimated snow or ice density to convert volume into mass changes, the two methods should obtain fairly similar numbers for the glacier wide mass balance. In this way, geodetically obtained results can be used to cross-check glaciological mass balances on various time-scales (Zemp et al., 2013, and references therein).

In the subsequent sections we introduce the glaciological and the geodetic measurement methods as applied at Hintereisferner. We first determine a common base for the two datasets, by the homogenization of glacier outlines and DEMs, followed by quantifying method-inherent uncertainties.

## 3.1 The glaciological method

Annual glaciological measurements at Hintereisferner commenced in 1952 (Hoinkes, 1970), resulting in one of the longest continuous glacier mass balance time series worldwide. The distribution of 40 to 50 (maximum 100) ablation stakes over the main tongue of Hintereisferner is a compromise between representative coverage and logistic feasibility (Kuhn et al., 1999; Fischer, 2011). During the study period no ablation stakes were maintained in the upper part of the glacier, where the accumulation was usually determined by means of snow pits and probings at the end of the mass balance year. The location of individual snow pits has been kept more or less constant over the whole study period. Their number changed according to the varying extent of the accumulation area from none in e.g. 2002/03 up to 14 pits in 2003/04 (see Fig. 1). The series follows the fixed date system as defined by the hydrological year, spanning from October $1^{st}$ to September $30^{th}$ of the following year, with additional measurements in spring and during about fortnightly visits between June and October.

The annual mass balance at each measurement point is derived by converting the individual change of surface height as obtained from stakes and pits. Ice ablation obtained from repeat stake readings is converted into point specific mass balance by applying an assumed constant density of $900\ \mathrm{kg\,m^{-3}}$. Accumulation is determined by measuring the snow depth in conjunction with depth-averaged snow density in snow pits. The point values and additional observational information such as the position of the snowline from an automatic camera and from terrestrial and air photographs, topographic conditions, and the expert knowledge about typical spatial patterns are the basis for drawing contour lines of equal mass balance. The resulting areas of equal mean mass balance are then intersected with 50 m altitude bands in order to derive the vertical mass balance profile. By integrating over the altitude bands, the total glaciological mass balance of the glacier $\Delta M_{glac}$ is obtained. Dividing $\Delta M_{glac}$ by the glacier area $S$ results in the glacier wide mean specific mass balance $B_{glac}$ (Cogley et al., 2011). Results are submitted to the World Glacier Monitoring Service (WGMS) annually (e.g. WGMS, 2015, 2012, and earlier volumes).

In order to provide a common base for both the glaciological and geodetic analyses we re-generate the annual glacier outlines from the ALS data strictly following the guidelines presented in (Abermann et al., 2010). The remaining annual random uncertainties due to possible errors in glacier outlines $\sigma_{glac.ref}$ are estimated as $\pm0.015\ \mathrm{m\,w.e.\,a^{-1}}$ (cf. Galos et al., 2017).

Before approaching the reanalysis of the annual surface mass balances of Hintereisferner for the time period 2001 to 2011 further uncertainties in the glaciological mass balances series must be addressed. The glaciological method suffers mainly from uncertainties related to (i) point measurements and (ii) their spatial extrapolation over the entire glacier (e.g., Zemp et al., 2013; Galos et al., 2017). Due to the lack of respective data on Hintereisferner we synthesize appropriate information from the literature to estimate both sources of uncertainty. Zemp et al. (2013) analysed the mass balance series of Hintereisferner for six periods between 1953 and 2006 and attributed an uncertainty of $\pm0.10\ \mathrm{m\,w.e.\,a^{-1}}$ to field measurements for the years after 1964 and doubled the value for the years before. For the spatial interpolation of point data they assigned values between $\pm0.14$ and $\pm0.54\ \mathrm{m\,w.e.\,a^{-1}}$ with an average of $\pm0.33\ \mathrm{m\,w.e.\,a^{-1}}$ for the entire period. Fountain and Vecchia (1999) found combined uncertainties for (i) and (ii) of up to $\pm0.33\ \mathrm{m\,w.e.\,a^{-1}}$ by analysing the modelled variability of the mass balance of South Cascade glacier. Thibert et al. (2008) and Thibert and Vincent (2009) analysed 51 years of mass balance for Glacier de Sarennes and reported a combined annual uncertainty of $\pm0.20\ \mathrm{m\,w.e.\,a^{-1}}$ for (i) and (ii). For Gries- and Silvrettagletscher,

Huss et al. (2009) assumed overall uncertainties related to (i) and (ii) of $\pm 0.16$ to $\pm 0.28$ m w.e. a$^{-1}$ . By investigating the glaciological and geodetic mass balances of Storglaciären, Zemp et al. (2010) determined the random uncertainty for (i) and (ii) with $\pm 0.10$ m w.e. a$^{-1}$ each, which resembles the results of Jansson (1999). For Findelengletscher, Sold et al. (2016) roughly estimated a random uncertainty of $\pm 0.04$ m w.e. a$^{-1}$ for (i), referring to Huss et al. (2009), and of $\pm 0.17$ m w.e. a$^{-1}$ for (ii) by

evaluating contour lines drawn independently by 18 analysers. On Nigardsbreen, Andreassen et al. (2016) obtained a total point measurement uncertainty of $\pm 0.25$ m w.e. a$^{-1}$ as the root sum square (RSS) of a false determination of the previous year's summer surface ($\pm 0.15$ m w.e. a$^{-1}$), up-welling of stakes ($\pm 0.20$m w.e. a$^{-1}$ ), and incorrect density assumptions of snow and firn ($\pm 0.05$ m w.e. a$^{-1}$). Uncertainty of spatial integration was taken as $\pm 0.21$ m w.e. a$^{-1}$, made up by point measurements insufficiently covering both the vertical range and the total area of the glacier.

Based on the findings of Zemp et al. (2013) combined with expert knowledge about the study site, we assess the uncertainty related to point measurements at Hintereisferner, to be in the order of $\sigma_{glac.point} = \pm 0.10$ m w.e. a$^{-1}$, resulting in a decadal value of about $\pm 0.32$ m w.e. For Hintereisferner we estimated the uncertainty related to extrapolation of point data based on Sold et al. (2016) leading to an annual value of $\pm 0.15$ m w.e. a$^{-1}$. Additionally we accounted for the presence of large areas not covered by point measurements. According to Andreassen et al. (2016), we assume that the extrapolation over those

areas inherits further uncertainties of $\pm 0.10$ m w.e. a$^{-1}$. Hence, the uncertainty due to spatial integration of the respective measurements over the entire glacier is defined to be $\sigma_{glac.spatial} = \pm 0.18$ m w.e. a$^{-1}$ and the related decadal uncertainty is $\pm 0.57$ m w.e.. Overall uncertainties for the glaciological mass balances are calculated, following Zemp et al. (2013, Eq. 14), leading to an annual value of $\sigma_{glac.total} = \pm 0.21$ m w.e. which corresponds to a cumulative uncertainty of the glaciological method (2001 to 2011) of $\pm 0.65$ m w.e.

**3.2  The geodetic method**

Between 2001 and 2011, 11 ALS flight campaigns had been carried out near the end of each mass balance year (see Tab. 1). During each ALS acquisition campaign, the glacier was covered with a number of overlapping flight strips in order to increase the point density and to ensure high quality and complete coverage of the glacier (Wever and Lindenberger, 1999; Geist et al., 2007). As there is essentially no high vegetation in the study area, ALS points are classified into ground points and flying

objects (outliers) only. The ground points of all datasets are imported into a laser database system (Rieg et al., 2014) which facilitates storage and further processing. DEMs of 1 m resolution were calculated for all datasets, whereby the mean value of all ALS points located in each cell represents the elevation of the cell. The elevation values for the few raster cells that do not contain a single point are interpolated from the neighbouring cells using a least squares method. In order to provide high-quality DEMs used for mass balance calculations, horizontal misalignment of the DEMs being differenced has to be excluded.

Therefore a statistical co-registration correction procedure as suggested by Nuth and Kääb (2011) was performed for this study. Following Joerg et al. (2012) we applied the first two steps of the procedure to the ice-free areas for identifying potential horizontal shifts and vertical offsets between two ALS-DEMs. The statistical co-registration reveals horizontal shifts smaller than the DEM pixel resolution with no elevation-dependent bias, hence, the DEMs can be subtracted from each other without performing DEM corrections. The total volume change $\Delta V$ between two dates is then derived from the respective elevation

difference $\Delta h_k$ of the two grids at pixel $k$ with cell size r of the DEMs, summed over the number of pixels $K$ covering the glacier and is expressed as follows (cf. Zemp et al., 2013):

$$\Delta V = r^2 \sum_{k=1}^{K} \Delta h_k. \tag{1}$$

For a comparison with the glaciological balance, $\Delta V$ is then converted into a specific mass balance in the unit meter water equivalent (m w.e.):

$$B_{geod} = \frac{\Delta V}{1/2 \cdot (S_{t0} + S_{t1})} \cdot \frac{\overline{\rho}}{\rho_{water}}, \tag{2}$$

where $S_{t0}$ and $S_{t1}$ are the glacier areas at the first ($t_0$) and second ($t_1$) acquisition date respectively and $\overline{\rho}/\rho_{water}$ is the ratio between the average bulk density (see Eq. 5 in Sect. 4.2) of $\Delta V$ and the density of water.

Despite a thorough co-registration, surface elevation differencing of two DEMs is still subject to various uncertainties. The vertical accuracy of the raw ALS point data was first assessed by comparing the point clouds with differential global navigation satellite system (dGNSS) measured points on a homogeneous horizontal surface outside the study area (in our case a football field in Zwieselstein 20 km down-valley of Hintereisferner). The standard deviations (SD) of vertical accuracies of the individual datasets are shown in Tab. 1. As the reference surface does not reflect the surface conditions in terms of slope, aspect and roughness, and therefore is not representative for vertical accuracies, Bollmann et al. (2011) compared dGNSS ground control points with laser returns (deviation to laser points 0.07 m, standard deviation 0.08 m) and calculated an absolute slope-dependent vertical accuracy for Hintereisferner ALS point data (<0.10 m on slopes <40°). Sailer et al. (2014) analysed the uncertainties resulting from rasterizing laser point clouds, revealing that a cell size of 1x1 m as used for our study causes only negligible errors of less than 0.10 m. For the geodetic balance ($B_{geod}$), the results of DEM differencing over stable terrain are taken to define uncertainties associated with the DEM comparison. Therefore, we selected five stable control areas ($3 \times 10^4 \, \mathrm{m}^2$) surrounding the glacier (Fig. 1), in order to quantify grid-based uncertainties of spatially averaged elevation differences. The selection of these sites is based on visual inspection and expert knowledge about the terrain around Hintereisferner (Bollmann et al., 2011; Sailer et al., 2012). According to (Rolstad et al., 2009), we assumed that the DEM uncertainty over stable terrain is representative for the entire glacier. However, we did not correct our sample size for spatial autocorrelation, but due to the high sampling density of the ALS-data used, we assumed that the number of independent items is about the number of glacier pixels (cf. Joerg et al., 2012). Thereby, the influence of random pixel-elevation uncertainty on the geodetic mass balance ($\sigma_{DEM}$) was calculated based on the stable control areas:

$$\sigma_{DEM} = \frac{SD_{\Delta Z}}{\sqrt{K}}, \tag{3}$$

where $SD_{\Delta Z}$ (see Tab. S1 in the Supplement) denotes the vertical standard deviation in stable control areas and $K$ is the total number of grid-cells used for the calculation of the glacier wide geodetic mass balance. This procedure yields uncertainties of $\pm 0.012 < \sigma_{DEM} < \pm 0.024$ m w.e., and $\sigma_{DEM} = \pm 0.087$ m w.e. for the 2001 to 2011 analysis (Tab. 3).

## 4 Method inherent differences

The differences between the glaciological and the geodetic mass balance series vary from year to year, being particularly high in certain years (Fig. 2, Tab. 2). The potential causes of these discrepancies are related to a number of factors: snow cover at the time of ALS acquisition, different glacier-wide density assumptions in mass balance calculation, survey date differences between the glaciological and geodetic observations, the way the methods consider the existence of crevasses and the different processes captured by the two mass balance methods. All those issues are thoroughly assessed below.

### 4.1 Differences induced by snow cover present in DEMs

Whereas the vertical accuracy of ALS-DEMs is high, biases as a result of snowfall events preceding the ALS surveys significantly influence the calculated volume change. From the analysis of elevation differences in the non-glaciated terrain, the mean difference between two DEMs in stable areas ($\overline{\Delta z}_{stable}$) can be used to correct for DEM-biases ($\epsilon_{DEM}$) caused by the presence of snow as follows:

$$\epsilon_{DEM} = \frac{\sum_{i=1}^{n} \overline{\Delta z_i}}{n}, \tag{4}$$

where n is the number of DEM grid cells covering stable and non-glacierized terrain. For the periods 2001/02, 2005/06, 2006/07 and 2007/08 the investigation of stable areas within the dDEMs revealed snow induced absolute vertical offsets between 0.18 and 0.58 m (see bold numbers for $\overline{\Delta z}_{stable}$ in table S1 of the Supplement). In all other dDEMs, the vertical bias was below 0.10 m. In 2004 and 2010 a snow fall event occurred some days before the ALS measurements. However, this is not reflected in the stable areas of the respective dDEM, because the snow in non glacierized areas had melted from off-glacier surface by the time of the ALS survey. This leads to a small offset in the non-glacierized terrain in the related mass balance periods. Yet, as snow cover increases, the ALS elevations measured on reference surfaces have to be cross-checked with snow depth data from the closest field survey and subsequently they have to be corrected. Based on the altitude distribution of stable areas and in-situ measurements a linear regression in 50 m elevation bands yields mean snow depths of 0.52 m in 2001, 0.23 m in 2004, 0.46 m in 2005, 0.13 m in 2006, 0.12 m in 2007 and 0.26 m in 2010. This leads to adjusted DEMs and, finally, to a respective mass balance correction value $\epsilon_{DEM}$ (Tab. 3). Furthermore this approach was integrated to the estimation of differences related to unequal survey dates (see Sect. 4.3).

### 4.2 Density conversion

While glaciological mass balances are derived calculating mass change based on well constrained in-situ measurements of density, geodetic balances are based on volume change measurements, which require volume-to-mass conversion using estimates of bulk density. Several studies assume that density in the accumulation area is constant over time and, hence, use glacier ice density for the conversion (e.g., Andreassen, 1999; Haug et al., 2009). But as long as snow or firn is present, doing so causes an overestimation of mass change. Hence, the use of the density of ice is only appropriate in glacier areas without firn. If year-to-year firn line changes are known, the volume-to-mass conversion can be improved by using an average density of firn

for changes in the accumulation area (e.g. Sapiano et al., 1998; Prinz et al., 2011).

In the present study, ice density ($\rho_{ice} = 900 \, \mathrm{kg \, m^{-3}}$) was only applied to the ablation areas, where altitudinal changes are either due to ice ablation or glacier dynamics while the geodetic mass change in (perennial) firn areas was calculated using a density of $\rho_{firn} = 700 \pm 50 \, \mathrm{kg \, m^{-3}}$ (Ambach and Eisner, 1966; Huss, 2013). Consequently, we calculate the annual conversion density $\overline{\rho}$ as used in Eq. 2 as follows:

$$\overline{\rho} = \frac{\rho_{ice} \cdot \Delta V_{ice} + \rho_{firn} \cdot \Delta V_{firn}}{\Delta V}, \tag{5}$$

where $\Delta V_{ice}$ and $\Delta V_{firn}$ are the volume changes in ice and firn areas respectively which both add up to the glacier wide volume change $\Delta V$.

In order to classify the glacier surface into ice and firn zones we designed a pixel-based surface classification work-flow based on ALS-intensity data following Höfle and Pfeifer (2007) and Fritzmann et al. (2011) (Fig. 3). This approach was applied to all years with suitable intensity-data while for years when no such data are available, the most contemporary ortho-images (2010) and/or LandsatTM images (2001 and 2004) were used for surface classification (see Fig. S1 in the Supplement). The resultant grids for each survey year were then used for a pixel-based conversion of volumetric changes to changes in mass. Respective values for the conversion density $\overline{\rho}$ lie in the range of 820 to 930 $\mathrm{kg \, m^{-3}}$ and are shown in Tab. 3. Although neither firn processes like compaction or melt water refreezing, nor the impact of glacier dynamics are explicitly resolved our approach is considered to notably improve the quality of our annual results compared to calculations based on a fixed glacier wide conversion density.

Uncertainties related to density conversion were estimated as follows: $\sigma_{dc}$ was assessed based on the estimated uncertainty ranges of $\rho_{ice}$ and $\rho_{firn}$ ($\pm 17$ and $\pm 50 \, \mathrm{kg \, m^{-3}}$) while $\epsilon_{dc}$ was calculated as the difference between our geodetic mass balance values and those based on a $\overline{\rho}$ of 850 $\mathrm{kg \, m^{-3}}$ as suggested by Huss (2013).

### 4.3 Survey date differences

Apart from 2011 with in situ measurements conducted on the same day as the ALS flight (Tab. S2 in the Supplement), the mass changes during the period between the survey dates of the two mass balance methods have to be considered. To align the geodetic dates with the end of the hydrological year used for the glaciological balances and for a corresponding adjustment of the geodetic results we incorporated data from in situ measurements and field work minutes as well as dDEM-based snow cover analysis (Sect. 4.1). Thereby ablation was assessed based on available stake readings during the late summer. Observed ablation trends between the observation dates were used to calculate mass change. If necessary, ablation was reconstructed by linearly extrapolating observed trends beyond the stake reading dates. Such cases were cross checked and adjusted based on meteorological data from Vent. The linear regression of point ablation versus altitude was finally used to calculate spatially extrapolated ablation. Note that the same altitudinal ablation gradient was used for the whole glacier since considerable ablation is restricted to the lower glacier part in this time of the year.

Accumulation between the ALS-survey and the fixed date was assessed based on recorded precipitation at Vent which was extrapolated to the glacier applying observed long-term precipitation gradients between Vent and 5 rain gauges in the Hintere-

isferner basin (Fig. 1). The snow-rain threshold of 0°C is obtained from the Vent temperatures along a lapse rate of 0.0065 $C\,m^{-1}$.

The survey date adjustment is performed individually for each annual geodetic mass balance, dependent on the presence/absence of snow during the field survey and the ALS data acquisition as well as on the difference between the survey dates and the end of the hydrological mass balance year. Accordingly, we proceeded as follows:

    i) If there was no snow cover during both surveys, and the ALS campaign took place before the field survey, an elevation dependent mean ablation gradient as described above is applied. This is the case in 2003 and 2008.

    ii) If there was no snow cover present during the field survey, but before a later ALS campaign, the mass balance has been adjusted to the survey date by subtracting the amount of snow from the corresponding DEM, as described in Sect. 4.1. This is the case for the years 2006 and 2007. The amount of snow determined for these years agrees well with extrapolated precipitation data from Vent.

    iii) If snow was present during the field survey, but the ALS campaign had been conducted before the snowfall event, the mass of the snow cover measured during the field survey is added to the geodetic mass balance using the measured densities and the linear regression of snow probings for the elevation distribution. This is the case in 2002 and 2008.

    iv) If snow was present during the field survey and the ALS data acquisition, the ALS-DEM was adjusted regarding the snow cover conditions. When the ALS campaign was conducted after the field survey, the geodetically determined snow height is subtracted (Sect. 4.1), and the mass of snow determined by field survey is added to the geodetic mass balance. This is the case for the years 2001, 2004, 2005, 2010.

Note that two corrections have been applied for the year 2008 when the ALS data acquisition took place 21 days before the field survey and ablation as well as accumulation occurred in this period. For 2009 and 2011 no survey date corrections were necessary due to ALS-measurements very close to September $30^{th}$.

## 4.4 Representation of crevasses

While crevasses are neglected in the glaciological method, they are partially resolved in the geodetic method. Although some crevasses might have been covered by snow during data acquisition, a number of big crevasses are visible in all DEMs. Depending on snow / melt conditions and their impact on ice movement, the recognition of crevasses in the single dDEMs and, hence, their impact on mass balance calculations varies widely. However, in this study we detected crevasses by assuming that they are deviations from a regular homogenous surface. By using the variance of elevation as a measure of terrain smoothness and by applying a closing filter, we derived a surface without crevasses (Kodde et al., 2007; Geist and Stötter, 2010). Consequently, we calculated the volume change of a "crevasse free" glacier, to quantify possible uncertainties due to open crevasses $\epsilon_{crev}$ in the geodetic mass balance (Tab. 3).

## 4.5 Internal and basal mass changes

Internal and basal mass balances are not captured by the glaciological method, but are implicitly included in the geodetic mass balances. Thus, when comparing glaciological with geodetic balances, internal and basal mass changes need to be assessed separately. Particularly for mountain glaciers studies on this topic are rare and published values represent estimates rather than verified measurements. On Storglaciären, for example, Östling and Hooke (1986) estimated the contribution of basal melt due to geothermal heat as about -0.001 m w.e. a$^{-1}$ and Holmlund (1987) suggested -0.01 m w.e. a$^{-1}$ of internal melt caused by the release of potential energy from run-off. Albrecht et al. (2000) considered internal ablation due to ice motion being small on Storglaciären and, thus, negligible. For South Cascade Glacier, Mayo (1992) estimated the combined effect of frictional/geothermal basal melt, melt by the release of potential energy of water and melt by the loss of potential energy through ice-flow as -0.09 m w.e. a$^{-1}$. Thibert et al. (2008) estimated -0.009 m w.e. a$^{-1}$ of basal ablation due to geothermal heat and -0.008 m w.e. a$^{-1}$ of internal melt due to water flow on Glacier de Sarennes over a period of 51 years. Huss et al. (2009) estimated the contribution to ablation of geothermal heat, internal deformation, and basal friction as -0.01 m w.e. a$^{-1}$ for glaciers in the Alps. Andreassen et al. (2016) calculated internal and basal ablation based on Oerlemans (2013) for 10 glaciers in Norway, yielding a range of -0.01 to -0.08 m w.e. a$^{-1}$. Sold et al. (2016) assessed a value of -0.014 m w.e. a$^{-1}$ for internal and basal processes at Findelengletscher following different previous studies (e.g. Herron and Langway, 1980; Pfeffer et al., 1991; Medici and Rybach, 1995; Huss, 2013).

In this study, we assess internal and basal ablation related to the dissipation of potential energy following Oerlemans (2013) and Andreassen et al. (2016). The resultant values are in the order of -0.04 m w.e. a$^{-1}$ which corresponds well to data for glaciers similar to Hintereisferner in terms of size and climate setting published by Oerlemans (2013). Melt from basal friction and geothermal heat flux was estimated according to Huss et al. (2009) as about -0.01 m w.e. a$^{-1}$. Hence, we estimate the total contribution of basal and internal processes to the mass balance to be -0.05 m w.e. a$^{-1}$.

## 5 Results

### 5.1 Glaciological mass balance

Within this study existing glaciological mass balance records were homogenized in terms of reference area (see Sect. 3.1) in order to make them comparable to the geodetic analyses. This showed only minor impact since glacier outlines have been frequently updated in the original record. However, the use of methodologically homogenized glacier outlines based on Abermann et al. (2010) changed the annual glaciological balances between -0.015 to +0.039 m w.e. a$^{-1}$ (see $\epsilon_{ref}$ in Tab. 2), while the over-all impact over the 2001 to 2011 period is +0.12 m w.e.. Numbers for annual glacier-wide specific mass balances range from -0.624±0.21 m w.e. in 2001/02 to -1.813±0.21 m w.e. in 2006/07. Results for individual years are shown in Fig. 2 and in Tab. 2 while the altitudinal profiles of glaciological mass balance are depicted in Fig. 4. Note that the uncertainty range $\sigma_{glac} = \pm 0.21$ m w.e. represents the random uncertainty as assessed in Sect. 3.1 and does not reflect any possible deficiencies in the glaciological series which shall be detected in the subsequent reanalysis.

## 5.2 Geodetic mass balance

The corrected geodetic mass balance of Hintereisferner over the ten years period 2001 to 2011 is -13.41±0.29 m w.e. which is 1.22 m w.e. more negative than the cumulative glaciological series (Tab. 2). Annual results range from -0.654±0.06 m w.e. in 2003/04 to -2.713±0.18 m w.e. in the year 2002/03 (Tab. 2).

5    The geodetic mass balance of Hintereisferner over the entire study period was mainly affected by snow being present in the year 2001 resulting in $\epsilon_{DEM}$ = +0.29 m w.e. Taking into account the effect of fresh snow on the DEMs of individual years (Sect. 4.1) leads to -0.41 < $\epsilon_{DEM}$ < +0.32 m w.e. The value of -0.41 m w.e. occurs in 2004/05 when snow was present at both ALS flight campaigns (Tab. 3) making up for 37% of the uncorrected mass change in this year.

Applying the work-flow for the spatially distributed density conversion (Sect. 4.2) leads to $-0.04 < \epsilon_{dc} < +0.31$ m w.e., with 10    maxima in 2002/03 and 2005/06 (Tab. 3). These maxima are due to the total lack of snow and firn at the end of these mass balance years. The uncertainty related to our density assumption (Sect. 4.2) lies between $\pm0.02 < \sigma_{dc} < \pm0.18$ m w.e. with ±0.27 m w.e. over the entire period of record.

Values for adjustments related to survey date correction are in the order of $-0.08 < \epsilon_{sd} < +0.06$ m w.e. (Sect. 4.3 and Tab. 3). Significant melt amounts between ALS flight and field survey dates occur on small parts of the glacier tongue only. Ice 15    ablation of almost 1 m at the lowest stakes of Hintereisferner measured between $30^{th}$ September (field survey) and $8^{th}$ October (ALS campaign) 2006 corresponds to a glacier wide specific mass loss of only 0.03 m w.e. during the same time. Uncertainties related to the consideration of crevasses ($\epsilon_{crev}$) in the geodetic method are small and vary between -0.04 and +0.06 m w.e. with +0.05 m w.e. for the 2001 to 2011 period (Sect. 4.4 and Tab. 3). While the glacier wide effect of internal mass changes on an annual basis is $\epsilon_{int} = 0.05$ m w.e. a$^{-1}$, it is 0.50 m w.e. on the decadal timescale (Sect. 4.5 and Tab. 3).

20    Annual totals for ($\epsilon_{geod}$) are in the range of -0.38 to +0.57 m w.e. while the random uncertainties for individual years are $\pm0.029 < \sigma_{geod} < \pm0.183$ m w.e. (Tab. 3. The geodetic balance calculated from the 2001 and 2011 DEMs yields $\epsilon_{geod}$ = +1.03 m w.e. and $\sigma_{geod} = \pm0.29$ m w.e. All numbers for the applied corrections and the single uncertainty sources ($\epsilon$ and $\sigma$) are summarized in Table 3 while the altitudinal profiles of the glaciological and geodetic mass balances for each year (2001/02 to 2010/11) are shown in Fig. 4.

25 ## 5.3 Methodological intercomparison

The comparison of annual glaciological to geodetic balances shows that all but three annual data pairs match satisfyingly within the assessed uncertainty ranges (Fig. 2). The largest differences ($\Delta B = B_{glac.hom} - B_{geod.corr}$) between the two methods occur in the balance years 2002/03 with $\Delta B$ = +0.92 m w.e. and 2005/06 with $\Delta B$ = +0.60 m w.e. respectively. In 2006/07 the difference between glaciological and geodetic method is -0.45 m w.e., which means the geodetic result is less negative than 30    the glaciological one. The difference for the whole study period is 1.31 m w.e.. In order to detect significant biases between the two methods we calculated the reduced discrepancies ($\delta$) as described by Zemp et al. (2013) as

$$\delta = \frac{\Delta B}{\sigma_{common}}, \tag{6}$$

where the common variance $\sigma_{common}$ (Tab. 2) is defined as the RSS of the method-inherent uncertainties ($\sqrt{\sigma_{glac}^2 + \sigma_{geod}^2}$). The more consistent the two methods, the closer $\delta$ is to zero and the null-hypothesis on the 95% confidence level ($H_{0_{95}}$) can be accepted. As $\delta$ falls within the 95% confidence interval ($\delta < 1.96$) for seven annual (all but 2002/03, 2005/06 and 2006/07) and the cumulative mass balance values, the two applied methods can be considered as statistically similar (Tab. 2). Note that this approach is mainly designed for comparisons on longer (typical 10 years) time scales since biases on the annual scale might be missed. Nevertheless, in our case it allows the identification of significant deviations in three years.

From the common variance it is also possible to calculate the smallest bias that could theoretically be detected in the glaciological record (Zemp et al., 2013). The bias calculated at the 5% risk limit lies between 0.75 and 0.99 m w.e. and is by far larger than the calculated uncertainty of annual glaciological balances of 0.21 m w.e. In contrast, the detectable bias decreases with the length of the analysed period, which can be explained by error propagation.

## 6    Discussion

In search for possible causes of these large discrepancies between the methods in three of the sampled years, we explore the potential contribution of individual components of $\epsilon_{geod}$ in the years of concern: The influence of temporary snow cover ($\epsilon_{DEM}$) on the geodetic mass balances is high but a thorough consideration in our study ensures that the results are within the 95% confidence interval. In contrast, the survey date differences show little effect. Concerning the conversion of glacier volume to mass changes, we used a new classification approach to derive a more accurate value of annual conversion density ($\overline{\rho}$). Calculated values for $\overline{\rho}$ are in the range of 820-930 $\mathrm{kg\,m^{-3}}$. This is in line with the glacier-wide value of 850±60 $\mathrm{kg\,m^{-3}}$ recommended by (Huss, 2013). Nevertheless, in 2010 $\overline{\rho}$ reaches 930 $\mathrm{kg\,m^{-3}}$, a value which at a first glance appears unrealistic. In this year opposite signs of elevation changes in the accumulation and ablation area compensate for each other, which results in a conversion factor which is higher than the density of ice. Such is possible in cases of (i) short observation periods (1-3 years), (ii) small volume changes, (iii) strong year to year changes in the vertical mass balance profiles, or combinations of these factors. Our approach accounts for year to year changes in the spatial extent and distribution of the snow/firn zones. Highest uncertainties arise in the years 2002/03 and 2005/06 when all snow from the previous winter melted entirely. As the uncertainty associated with density is of particular importance (Moholdt et al., 2010; Huss, 2013) we conducted a sensitivity test for the periods of good agreement by holding all other parameters fixed. Densities calculated within our $\overline{\rho}$-range (Tab. 3) still lead to results within the 95% confidence interval.

As crevasses may influence geodetically calculated volume changes we assessed their impact on the geodetic method. The largest impact (0.06 m w.e., or 3% of glaciological mass balance) was detected for 2002/03 when numerous crevasses opened due to the extremely hot summer causing extraordinary high glacier velocities (Geist et al., 2007). Hence, crevasses contribute negligibly to the differences between geodetic and glaciological mass balances.

Internal and basal processes are also of rather minor importance (-0.05 m w.e. $\mathrm{a^{-1}}$; Sect. 4.5) and do not change the differences between the two data series substantially. Yet, we note that in years with extreme melt rates as in 2003 and 2006 additional melt water from outside the glacier may enter the glacier bed in the tongue-area during the ablation season which leads to basal melt

rates possibly exceeding the above estimate. However, even a doubling of our estimate to -0.10 $\mathrm{m\,w.e.\,a^{-1}}$ does not explain the large discrepancies between the glaciological and geodetic method in the years 2002/03, 2005/06 and 2006/07.

Other uncertainties possibly contributing to the high mass balance discrepancies in 2002/03, 2005/06 and 2006/07 may be method-inherent uncertainties related to the field measurements, such as the false determination of the last year's summer sur-

face. This might be an issue for the high discrepancies in the individual survey years, but cannot be quantified due to the lack of corresponding information. However, none of the discussed issues can explain the high deviations between glaciological and geodetic analyses in the mass balance years 2002/03, 2005/06 and 2006/07.

Nevertheless, a potential reason is indicated by the altitudinal distribution of point measurements as shown in Fig. 5 for the exemplary year 2002/03. In all three of the poorly matched years, glaciological point data from elevations above 3000 m a.s.l.

are missing on Hintereisferner (Fig. 6). Given the glacier-median elevation of about 3039 m a.s.l. this means that the upper half of the glacier was not covered by measurements in these years. At the same time the three years of concern are those with the most negative mass balances within the Hintereisferner record (Fig. 2 and Tab. 2). The reason for missing measurements in higher elevated areas in those years is the fact that no snow from the previous winter survived the warm summers at snow pit locations and hence, traditional accumulation measurements were not possible. To address the problem of a mass balance net-

work which had not been adapted in time, ablation rates measured at the highest stakes on the flat tongue (at about 3000 m a.s.l. and lower) had been multiplied with the observed ice exposure time of the higher slopes (G. Markl, personal communication). This disregards the impact of higher solar radiation intensity on the slopes compared to the flat tongue, and the application of formerly observed "typical" spatial patterns of mass balance in the spatial extrapolations are considered to be possible reasons for the differences between the two methods in these years.

After several years of gradual degradation of the firn body, ice and older dark firn had suddenly become exposed over all altitude bands by August 2003 with consequent effects on albedo and the surface energy budget. The East and South facing high slopes of Hintereisferner exposed a low-albedo surface to high solar radiation for several weeks in the exceptionally warm and dry summer 2003 (Fink et al., 2004). As a consequence, the mass loss in the former accumulation area of Hintereisferner became unexpectedly large in areas without ablation stakes ($> 50\%$ of the glacier area). As a consequence, well known spatial

patterns of surface melt of former years used in the mass balance analyses were no longer valid; an effect which had also been observed on a smaller glacier in the Eastern Alps some years earlier (Kaser et al., 2001).

While higher winter snow cover buried the dark ice surface far enough into the autumns of 2004 and 2005 protecting higher glacier portions and allowing for snow pits at the end of summer, the 2002/03 problem became evident again in summer 2006 when dark glacier surfaces were again exposed after an early-summer heat wave.

In 2006/07 when the glaciological mass balance obtains more negative values than the geodetic one we face a different situation. During summer 2007 a number of snow fall events increased the surface albedo in the upper part of Hintereisferner while stake measurements in the lower part of the glacier indicated relatively high ablation rates. We suspect that those high ablation rates were mistakenly extrapolated to higher elevations but the lack of meta-data for this particular year disables any further discussion and interpretation.

However, based on our findings we argue for the geodetic data being closer to reality than the glaciological ones in the years

2002/03, 2005/06 and 2006/07 (cf. Thibert et al., 2008; Huss et al., 2009). For all other years when differences between the methods are statistically insignificant and where error bars overlap the glaciological analyses yield plausible results. This interpretation is corroborated by comparison of the mass balance of Hintereisferner with those of other glaciers in the region (see Fig. S3 in the supplementary material).

## 7   Conclusions

Over the past decades it has become a standard procedure to review annual glaciological data alongside with decadal geodetic mass balances from a variety of sources (e.g., Kuhn et al., 1999; Hagg et al., 2004; Cox and March, 2004; Thibert et al., 2008; Huss et al., 2009; Fischer, 2011; Galos et al., 2017). However, none of the mentioned studies uses annually obtained high-resolution ALS-data over one decade. Geist et al. (2007) were the first authors comparing glaciological and ALS-based geodetic results on an annual time scale at Hintereisferner for the period 2001 to 2005. Their findings revealed considerable differences between the methods, especially in the year 2002/03. Yet, the study focuses on methical issues only and neither includes a thorough data homogenisation nor a robust uncertainty assessment and discussion.

In our review of the 2001 to 2011 Hintereisferner mass-balance record we show that the explicit consideration of uncertainty sources, such as the presence of snow cover, survey dates and density assumptions, is mandatory for accurately calculating annual geodetic mass balances. Conversely, crevasses and internal processes seem not to play a key role. The largest potential source for differences between the geodetic and glaciological method on the annual scale is the presence of a snow cover during geodetic data acquisition. Although its reliance on a variety of raw data and meta information might limit its applicability to other sites or cases, our method allows correction for method-inherent differences and provides an appropriate basis for detecting discrepancies in the direct glaciological method. Joint analysis of glaciological and geodetic data series shows that the glaciological method in our case successfully captures the mass change in seven out of ten mass balance years and both methods generally agree on the annual as well as on the decadal time scale.

Our analysis further shows that in years with very negative mass balances and a low extent of the accumulation area, the glaciological measurement network has to be adapted accordingly. In the case of Hintereisferner, this means that additional ablation stakes in higher parts of the glacier are needed to properly assess the mass changes in regions where snow measurements could be performed in former times. If appropriate changes to the measurement network are not made in time, attempting to overcome the resultant lack of data with mass balance extrapolation approaches based on spatial patterns observed during preceding years, might be inappropriate. In the 2001 to 2011 Hintereisferner series the application of such approaches led to considerable deviations from the geodetic results in three years and the careful revision of both series identifies three cases where the applied glaciological measurement set-up proves deficient. Hence, we conclude that in times of increasing availability of high resolution topographic data, geodetic mass balances can represent a valuable possibility to unravel shortcomings in the glaciological measurements even on an annual scale if these data are thoroughly analysed.

Although major discrepancies between the glaciological and geodetic methods on Hintereisferner could be explained by our

work-flow, further investigations should address a better quantification of error sources, such as internal and basal processes, in both the glaciological as well as geodetic mass balances. Moreover, in times of vanishing firn areas and disconnecting glacier tributaries, existing mass balance measurement networks might have to be reassessed.

With the high-quality ALS-DEMs reliably reproducing the annual mass balance the work-flow presented here is recommended

for (i) a re-analysis of annual glaciological with annual geodetic data and (ii) as a grid-based tool for deriving a glacier-wide geodetic mass balance of high spatial resolution suitable for a better understanding of the nature and origin of the differences between the two methods.

*Data availability.*    Mass balance data related to this study are submitted to the WGMS and will hence be publicly available through their website. Additional information on study site and data are available on request at the Institute of Geography (ALS and geodetic data) and the

Institute of Atmospheric and Cryospheric Science (glaciological and meteorological data), University of Innsbruck. Coarser (10 m) versions of all the ALS-DEMs are available at Pangea.de (https://doi.pangaea.de/10.1594/PANGAEA.875889).

*Author contributions.*    CK performed the gross part of analyses, designed and wrote the paper, EB designed an early version of the study and carried out preliminary analyses, SG contributed to methical development and writing and led the revision-process of the paper, LN refined the paper by native-speaker editing and comments on the manuscript, RP added expert knowledge on study site and data and

performed related analyses, LR was involved in handling and analysing ALS-data, RS, JS and GK are the leaders of scientific projects related to this study and contributed to study-related discussions, paper design and writing. All authors contributed to the refinement of the manuscript.

*Competing interests.*    The authors declare that no competing interests are present

*Acknowledgements.*    The ALS data are hosted at the Institute of Geography of the University of Innsbruck and were acquired during differ-

ent scientific projects: The EU-project OMEGA – Operational Monitoring of European Glacial Areas (project-no.: EVK2-CT-200-00069), the asap – Austrian Space Applications Programme ALS-X (project-no.: 815527), the ACRP – Austrian Climate Research Programmes C4AUSTRIA (project-no.: A963633). Mass balance data, field survey minutes and meteorological data are provided from the Institute of Atmospheric and Cryospheric Sciences of the University of Innsbruck. Glaciological mass balance acquisition is funded by the Tyrolean Hydrological Survey (Hydrographischer Dienst Tirol).

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

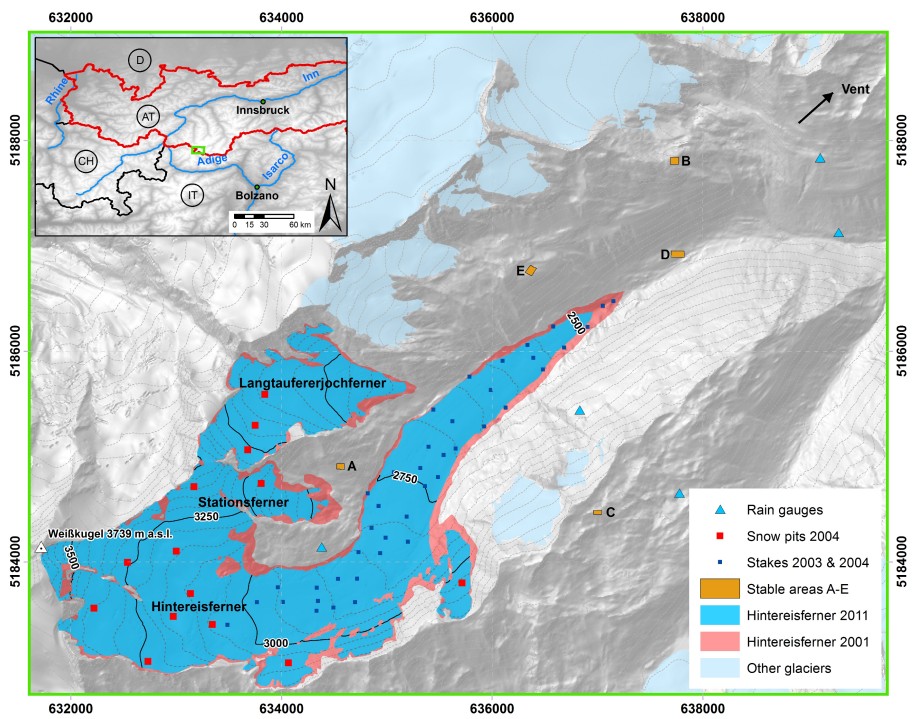

**Figure 1.** A map of Hintereisferner with the locations of the rain gauges and the glaciological mass balance measurement points in 2004 as an example. Also depicted are the glacier outlines for 2001 and 2011. Note that in 2003 no accumulation measurements could have been carried out due to the strongly reduced accumulation zone. Hence, only ablation stakes were available. Coordinates are in WGS84/UTM32N.

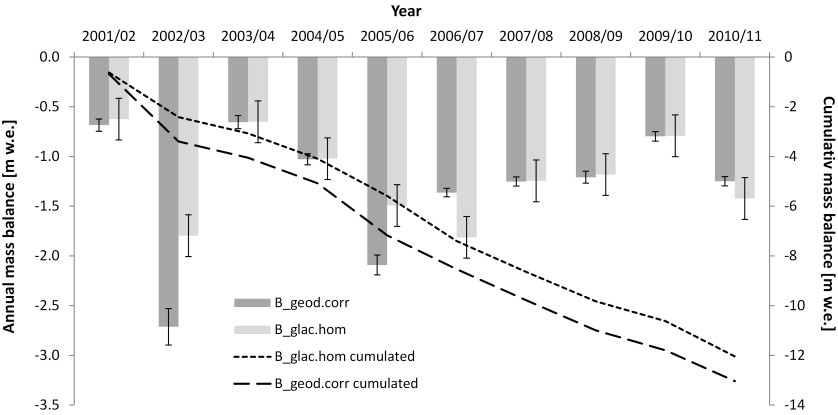

**Figure 2.** Comparison of the homogenized glaciological and corrected geodetic annual and cumulative mass balances of Hintereisferner over the study period. Dark grey bars and the dashed black line indicate geodetic balances while light grey bars and the dotted black line show the glaciological series. Vertical black lines show the annual uncertainties ($\sigma_{glac}$ and $\sigma_{geod}$) of the two methods.

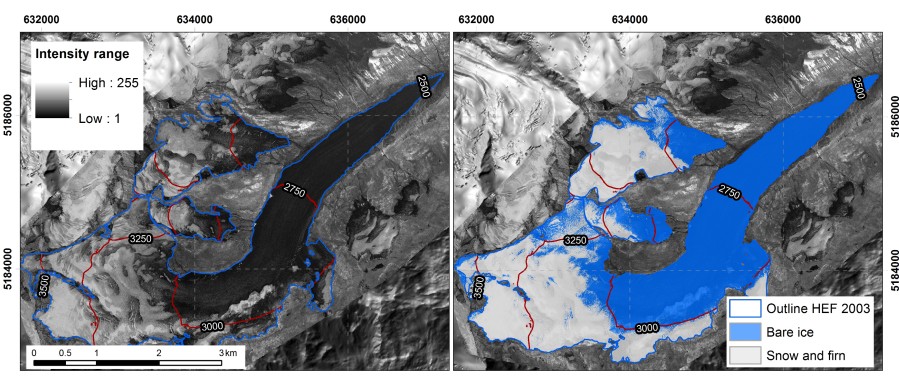

**Figure 3.** Intensity of the reflected laser beam of the ALS acquisition in 2008 (left) and derived surface classes (right). The classes are perennial firn with an average density of $700 \pm 50\ \mathrm{kg\,m^{-3}}$ and bare glacier ice of $900 \pm 17\ \mathrm{kg\,m^{-3}}$. Map coordinates are in WGS84/UTM32N.

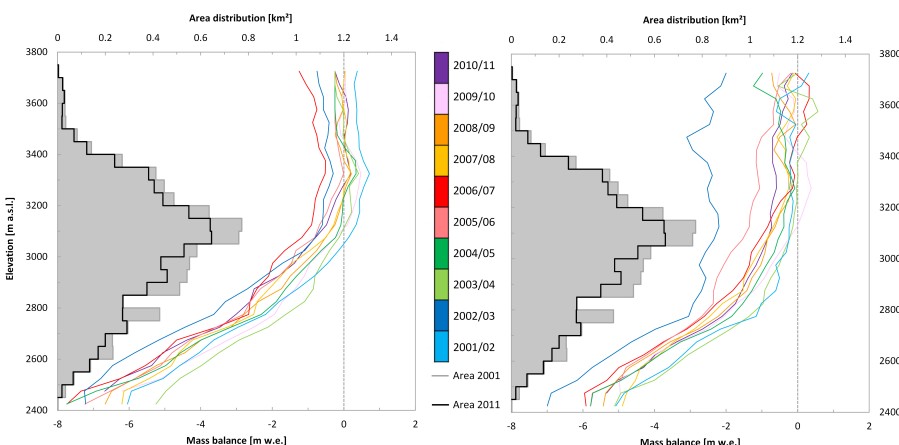

**Figure 4.** Altitudinal profiles of annual homogenized glaciological (left) and geodetic (right) mass balances over the study period. Note that vertical profiles of the two methods cannot be directly compared due to the effect of glacier dynamics not captured in the glaciological results.

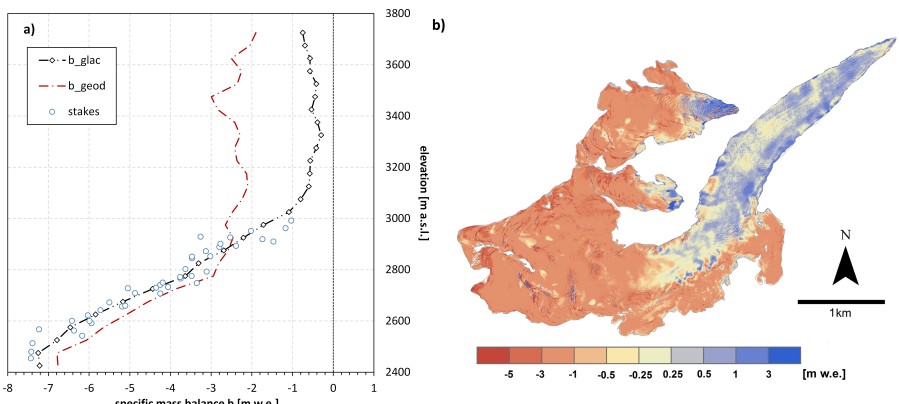

**Figure 5.** The extraordinary mass balance year 2002/03. (a) Comparison of vertical mass balance profiles ($B_{glac}$; $B_{geod}$) including the distribution of direct measurement points over the elevation-span of the glacier. (b) Spatially distributed difference of the methodical results with main deviations between the methods in elevations higher than 3000 m a.s.l. where in situ observations are missing. Note that vertical profiles of the two methods cannot be directly compared due to the effect of glacier dynamics which leads to more negative geodetic results (than the glaciological ones) in the higher elevated areas and vice versa in the lower glacier regions.

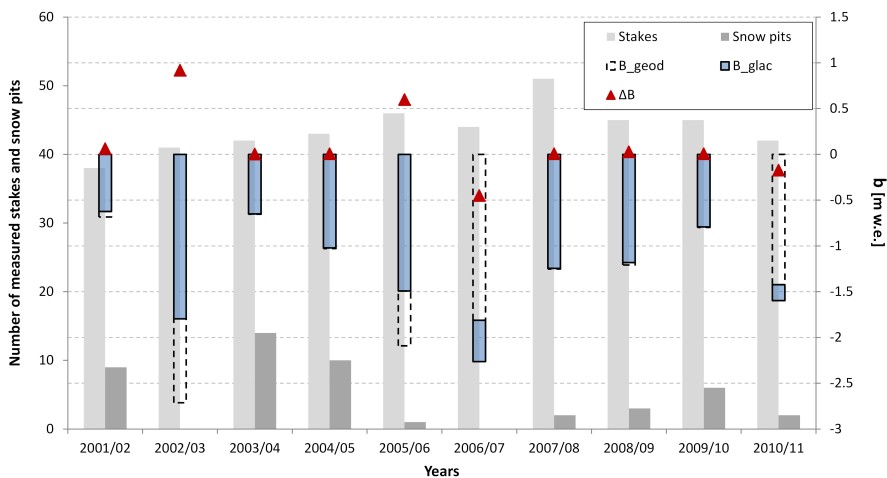

**Figure 6.** Comparison of mass balances ($B_{glac.hom}$ and $B_{geod.corr}$) and their differences ($\Delta B$) with number of ablation and accumulation measurements. Note that in areas higher than 3000 m a.s.l. only accumulation measurements were performed.

**Table 1.** Key parameters for the 11 ALS data acquisition campaigns at Hintereisferner from 2001 to 2011. Point density is averaged over the study area, while the horizontal accuracy is calculated based on a flat reference area in the vicinity of the study area.

| Acquisition date | Optech sensor | Mean height above ground | Max scanning angle | Pulse repetition frequency | Across track overlap | Average point density | Vertical accuracy (standard deviation) |
|---|---|---|---|---|---|---|---|
| | | $[m]$ | $[degrees]$ | $[Hz]$ | $[\%]$ | $[points\ per\ m^2]$ | $[m]$ |
| 11.10.2001 | ALTM1225 | 900 | 20 | 25000 | 24 | 1.1 | n.a. |
| 18.09.2002 | ALTM3033 | 900 | 20 | 33000 | 24 | 1 | 0.1 |
| 26.09.2003 | ALTM1225 | 900 | 20 | 25000 | 24 | 1 | 0.06 |
| 05.10.2004 | ALTM2050 | 1000 | 20 | 50000 | 24 | 2 | 0.07 |
| 12.10.2005 | ALTM3100 | 1000 | 22 | 70000 | 50-75 | 3.4 | 0.07 |
| 08.10.2006 | ALTM3100 | 1000 | 20 | 70000 | 37-75 | 2 | 0.08 |
| 11.10.2007 | ALTM3100 | 1000 | 20 | 70000 | 37-75 | 3.4 | 0.06 |
| 09.09.2008 | ALTM3100 | 1000 | 20 | 70000 | 40-45 | 2.2 | 0.06 |
| 30.09.2009 | ALTM3100 | 1100 | 20 | 70000 | 31-66 | 2.7 | 0.05 |
| 08.10.2010 | ALTM Gemini | 1000 | 25 | 70000 | 62 | 3.6 | 0.03 |
| 04.10.2011 | ALTM3100 | 1100 | 20 | 70000 | 25-75 | 2.9 | 0.04 |

**Table 2.** Original glaciological mass balances ($B_{WGMS}$), the impact of reference-area adjustment ($\epsilon_{ref}$), the homogenized glaciological mass balance $B_{glac.hom}$ with related random uncertainties $\sigma_{glac}$, the corrected geodetic mass balances $B_{geod.corr}$ and their uncertainties $\sigma_{geod.corr}$, the difference between homogenized glaciological and corrected geodetic balances $\Delta B$, the common variance of the two series $\sigma_{common}$ and the reduced discrepancies $\delta$. The acceptance of the null-hypothesis ($H_{0_{95}}$), indicating if the glaciological balance is statistically different from the geodetic balance or not, is evaluated on the 95% confidence level, which corresponds to $\delta$-values inside (outside) the $\pm 1.96$ range, respectively. $\beta_{95}$ depicts the probability of fulfilling $H_{0_{95}}$ inspite of differences at the 95% confidence level. Bold entries refer to years in which $H_{0_{95}}$ is not fulfilled.

| Period | $B_{WGMS}$ | $\epsilon_{ref}$ | $B_{glac.hom} \pm \sigma_{glac}$ | $B_{geod.corr} \pm \sigma_{geod.corr}$ | $\Delta B$ | $\sigma_{common}$ | $\delta$ | $H_{0_{95}}$ | $\beta_{95}$ |
|---|---|---|---|---|---|---|---|---|---|
| 2001/02 | -0.647 | +0.023 | -0.624 ±0.21 | -0.685 ±0.062 | 0.061 | 0.215 | 0.28 | yes | 94 |
| **2002/03** | **-1.814** | **+0.018** | **-1.796 ±0.21** | **-2.713 ±0.183** | **0.917** | **0.276** | **3.33** | **no** | **9** |
| 2003/04 | -0.667 | +0.016 | -0.651 ±0.21 | -0.654 ±0.063 | 0.003 | 0.216 | 0.01 | yes | 95 |
| 2004/05 | -1.061 | +0.039 | -1.022 ±0.21 | -1.028 ±0.056 | 0.006 | 0.214 | 0.03 | yes | 95 |
| **2005/06** | **-1.516** | **+0.023** | **-1.493 ±0.21** | **-2.091 ±0.100** | **0.598** | **0.229** | **2.61** | **no** | **26** |
| **2006/07** | **-1.798** | **+0.015** | **-1.813 ±0.21** | **-1.363 ±0.041** | **-0.450** | **0.210** | **-2.14** | **no** | **43** |
| 2007/08 | -1.235 | +0.011 | -1.246 ±0.21 | -1.252 ±0.046 | 0.006 | 0.211 | 0.03 | yes | 95 |
| 2008/09 | -1.182 | +0.000 | -1.182 ±0.21 | -1.209 ±0.060 | 0.027 | 0.214 | 0.14 | yes | 95 |
| 2009/10 | -0.819 | +0.027 | -0.792 ±0.21 | -0.808 ±0.029 | 0.016 | 0.208 | 0.08 | yes | 95 |
| 2010/11 | -1.420 | +0.003 | -1.423 ±0.21 | -1.249 ±0.047 | -0.174 | 0.211 | -0.82 | yes | 87 |
| 2001/11 | -12.307 | +0.117 | -12.190 ±0.65 | -13.413 ±0.291 | 1.223 | 0.714 | 1.71 | yes | 59 |

**Table 3.** Method-inherent differences and uncertainties as quantified in this study. Differences related to DEM ($\epsilon_{DEM}$ and $\sigma_{DEM}$), density conversion ($\epsilon_{dc}$ and $\sigma_{dc}$), survey dates ($\epsilon_{sd}$), internal processes ($\epsilon_{int}$ and $\sigma_{int}$) and crevasse volume ($\epsilon_{crev}$). While the overall $\epsilon_{geod}$ accumulates from all individual differences, the overall $\sigma_{geod}$ is calculated by propagating the individual uncertainties. The unit for $\overline{\rho}$ is $\mathrm{kg\,m^{-3}}$. All mass balance uncertainties are given in meter water equivalent (m w.e.).

| year | $\overline{\rho}$ | $\epsilon_{DEM}$ | $\epsilon_{dc}$ | $\epsilon_{sd}$ | $\epsilon_{int}$ | $\epsilon_{crev}$ | $\epsilon_{geod}$ | $\sigma_{DEM}$ | $\sigma_{dc}$ | $\sigma_{int}$ | $\sigma_{geod}$ |
|------|------|------|------|------|------|------|------|------|------|------|------|
| 01/02 | 830±30 | +0.289 | +0.081 | -0.032 | +0.05 | -0.024 | +0.364 | ±0.015 | ±0.058 | ±15 | ±0.062 |
| 02/03 | 820±45 | +0.090 | +0.309 | +0.061 | +0.05 | +0.062 | +0.572 | ±0.024 | ±0.181 | ±15 | ±0.183 |
| 03/04 | 875±20 | -0.197 | +0.013 | +0.026 | +0.05 | -0.037 | -0.145 | ±0.008 | ±0.061 | ±15 | ±0.063 |
| 04/05 | 855±30 | -0.405 | +0.034 | -0.019 | +0.05 | -0.038 | -0.378 | ±0.012 | ±0.053 | ±15 | ±0.056 |
| 05/06 | 850±35 | +0.293 | +0.140 | -0.082 | +0.05 | -0.005 | +0.396 | ±0.013 | ±0.098 | ±15 | ±0.100 |
| 06/07 | 885±20 | -0.016 | +0.023 | -0.020 | +0.05 | +0.004 | +0.041 | ±0.004 | ±0.038 | ±15 | ±0.041 |
| 07/08 | 865±25 | +0.097 | +0.052 | -0.063 | +0.05 | -0.019 | +0.117 | ±0.023 | ±0.037 | ±15 | ±0.046 |
| 08/09 | 890±20 | -0.045 | +0.011 | -0.050 | +0.05 | -0.034 | -0.068 | ±0.014 | ±0.056 | ±15 | ±0.060 |
| 09/10 | 930±20 | -0.321 | -0.037 | -0.028 | +0.05 | -0.042 | -0.378 | ±0.008 | ±0.023 | ±15 | ±0.029 |
| 10/11 | 870±25 | +0.321 | +0.049 | +0.028 | +0.05 | -0.019 | +0.429 | ±0.016 | ±0.042 | ±15 | ±0.047 |
| 01/11 | 890±20 | +0.289 | +0.267 | -0.070 | +0.50 | +0.047 | +1.033 | ±0.087 | ±0.274 | ±0.047 | ±0.291 |