# Peer review of "Geodetic reanalysis of annual glaciological mass balances (2001-2011) of Hintereisferner, Austria"

_The Cryosphere, 2017_

## Referee Comment (RC1) · M. Zemp (Referee) · 30 Aug 2017

Christoph Klug and colleagues present a detailed reanalysis of annual glaciological and annual geodetic balances at Hintereisferner, Austria, obtained between 2001 and 2011. This study puts an airborne laser scanner (ALS) dataset with exceptional spatial and temporal resolution over an entire decade at its full value. The comparison of these geodetic results with the glaciological balances from an extensive in-situ network have been long overdue but are now carried out in a very thorough way and including an error assessment according to best international practises.

[Figure]

Hence, I can recommend the paper for publication in The Cryosphere after consideration of the following two substantial points and a list of suggestions for minor corrections and clarifications:

(a) DTM-related random uncertainty of geodetic balances: The authors use the standard deviation of the DTM-differencing over selected stable terrain as random uncertainty for the geodetic balance (cf. equation 3, lines 196-207). I do not agree with this approach because it assigns a local DTM error to a zonal glacier change value. The standard deviation of the elevation differences on stable terrain indicates the uncertainty of the DTM differences for individual pixels. Instead, I propose to use the standard error, defined as the standard deviation divided by the square root of the number of independent items of information in the sample (cf. Zemp et al. 2013, The Cryosphere, Section 2.3). In the present case of ALS (> 1 point per m2) it can probably be assumed that the number of independent items is about the number of glacier pixels (cf. Joerg et al. 2012, RSE). Note that there is also the implicit assumption that the DTM uncertainty over stable terrain is representative for the DTM uncertainty over the glacier (cf. Rolstad et al. 2009, J. Glaciol.). Maybe that needs just to be mentioned somewhere in the paper.

(b) Geodetic method as substitution for the glaciological method: The authors conclude that the geodetic method (i) "can represent a valuable possibility to overcome shortcoming in the glaciological measurements even on an annual scale" (Lines 469-470) or (ii) "even as a substitute for the glaciological method". I can only partly support these conclusions for three reasons: (1) the geodetic and the glaciological methods are rather complementary in nature (than to substitute each other): the strength of the glaciological method is to capture the spatial and temporal variability of the glacier surface balance even with only a small sample of observation points but it is sensitive to systematic errors which accumulate linearly with the number of seasonal or annual measurements. The geodetic balance is able to cover the entire glacier but requires a density conversion, which becomes more challenging over short time periods because

of meteorological influences on the elevation change. (2) the nature of uncertainties: typically, ten years of data are required for the detectable difference to become lower than the annual random "noise" of the glaciological balance (cf. Zemp et al. 2013, The Cryosphere). A validation at annual time intervals might actually miss a bias. (3) cost-benefit considerations: the costs of the geodetic method are one to two orders of magnitudes higher than the costs of the glaciological method.

I suggest adding a short section that discusses these issues and rewording the corresponding conclusions.

(c) List of suggestions for minor corrections and clarifications:

Page 3, Line 67, "first use of annual geodetic records": At South Cascade Glacier, annual results from both geodetic and glaciological methods have been analysed by Krimmel (1999): Robert M. Krimmel (1999) Analysis of difference between direct and geodetic mass balance measurements at south cascade glacier, washington, Geografiska Annaler: Series A, Physical Geography, 81:4, 653-658.

P4, L119, "Results are submitted to the WGMS...": you could add a reference to WGMS (2017, and earlier reports): WGMS 2017. Global Glacier Change Bulletin No. 2 (2014–2015). Zemp, M., Nussbaumer, S. U., Gärtner- Roer, I., Huber, J., Machguth, H., Paul, F., and Hoelzle, M. (eds.), ICSU(WDS)/IUGG(IACS)/UNEP/UNESCO/WMO, World Glacier Monitoring Service, Zurich, Switzerland.

P6, L178-184, Equation 2: the geodetic balance is usually calculated using the average glacier area of the two surveys (cf. Zemp et al. 2013, The Cryosphere, Eq. (5) and (6)). At annual time steps, this might not make a big difference, but for the decadal period with a surface area reduction of 15% it does become relevant.

P6, L188 & Fig 1, stable areas: I fully support the decision to complement the down-valley soccer field with stable areas near the glacier. Please add a short comment about the selection criteria for the stable areas A-E.
P8/9, L240-267, density conversion: the density conversion factor depends on changes in the three-dimensional firn body and is a function of (i) the additional snow layer incl. related densification and metamorphosis, (ii) firn compaction and metamorphosis, and (iii) sub/emergence velocity. From the text, I cannot fully comprehend how these factors are covered (or not) by the author's approach combining differential DTMs, surface classifications, and density assumptions. Please clarify and discuss the opportunities and limitations of the used approached.

P9, L266-267 & Table 5, density conversion factor and related uncertainties: for a non-expert it is hard to follow how the density conversion factor and corresponding random uncertainties (together with the annual balance) relate to K.sigma and K.epsilon in Table 5. Adding a corresponding equation in Section 4.2 might help.

P9, L271, "stratigraphic year": I think this should be "end of the hydrological year" or "fixed date system" (cf. P9, L275, "30th September").

P10, L285-287, "elevation dependent mean ablation gradient": do you use the same gradient for the ablation and the accumulation zone? Please clarify.

P11, L323-324: for comparability, convert the values by Thibert et al. (2008) to annual change rates.

P15, L448-251, "were the first and so far only": consider rewording in view of earlier studies at South Cascade by Krimmel (1999, Geogr. Ann.).

Text, Figs & Tabs, "altitude" versus "elevation": in most cases, you could replace "altitude" by "elevation" (cf. McVicar, T. R., & Körner, C. (2013). On the use of elevation, altitude, and height in the ecological and climatological literature. Oecologia, 171(2), 335-337.)

P24, Fig. 1: For clarification, you could write in the figure caption: "Note that in 2003, no accumulation measurements COULD have been carried out DUE TO THE STRONGLY REDUCED ACCUMULATION ZONE. HENCE, only ablation stakes were available."

P25&30, Fig. 2 & 7: the two figures are redundant to a certain degree. On the other side, it is not fully clear, which differences and uncertainties are included. Please at least clarify in captions. In addition, you could consider merging Fig 2 & 7, showing bias corrections for both glaciological and geodetic results. Instead, you could remove the cumulative curves (=> shown in Fig. 8).

P26. Fig. 3: I would add a bar showing the intensity range (values) to the legend of the left image. In the legend of the right one, I would replace "perennial firn" by "snow and firn".

P27, Fig. 4: In the caption, please clarify what you mean with "Corrected". It might be sufficient adding a reference to the corresponding section in the paper. I would add the terms glaciolocial and geodetic to the label of the x-axis in the left and right figure, respectively. In addition, please add a note on the effect of the sub/emergence velocity.

P28, Fig. 5: you could add the data point(s) for the full period (glaciol.cum versus geod.cum, glaciol.cum versus geod.01/11).

P29, Fig. 6: please add a note on the effect of the sub/emergence velocity.

P31, Fig. 8: typically, one would calibrate the glaciological with the geodetic over the decadal period (i.e. 2001-11). Hence, it might be good to show that result here too.

P34, Tab. 2: you could add a column for the two dDTM of the full period, i.e. 01/11.

P35, Tab 3: please explain why the density given in the cation (900 kg m-3) differs with the one mentioned in the text (850 kg m-3, cf. P8, L249)

P36, Tab 4: in the caption, there are some problems with the symbol for average SC. What is the "mean acc. area"? Do you refer to the end-of-summer accumulation area?

P37, Tab 5: I would expect the annual uncertainties for the density conversion (sigma K) to be larger than for the (zonal) ones for the ALS-DTM (sigma DTM)... see also my comments above (substantial point (a) and comment related to density conversion, P9,

L266-267).

P38, Tab 6, caption: consider rewording "improved balance" into "bias-corrected balances"; consider rewording "statistical significance" by "reduced discrepancy". Use the same symbol for the common variance in caption (now wrongly epsilon.comvar) and table (=> sigma.comvar).

---

## Referee Comment (RC2) · Anonymous Referee #2 · 14 Sep 2017

Summary:

Klug and coauthors compare annually-resolved geodetic (airborne laser altimetry) and traditional mass balance records of Hintereisferner Glacier over the period 2001-2010. They find that for most years these two methods estimate similar mass change for the glaciers (within uncertainties). They note discrepancy between the methods for three years and attribute these differences to errors in the traditional mass balance data.

I found this paper (and their experimental design) to be well thought out and mostly

well described. It should be published. Globally, the number of traditional mass balance records relative to the number of existing glaciers is vanishingly small. We thus require alternative approaches that could complement conventional methods of measuring surface mass balance. Klug and coauthors make a convincing case that estimating mass balance using geodetic techniques is appropriate and in some cases can identify errors in the traditional series. This is especially important when one uses these series for calibration of models (hydrologic and mass balance).

I would recommend that the paper be considered for publication, but only after the authors make a number of substantive changes. Most of these suggestions are minor, but several will require some thought and future analysis. Below, I expand on the major points the authors should address. In several places I found the writing to be muddled and fraught with logic gaps and grammatical/typographical errors that might arise if English was not the first language of the senior author. I would encourage all authors spend time going through the revised manuscript to ensure its presentation is up to the standards required for the journal.

General Comments:

A) Streamline introduction - I found the introduction of the paper to be too long and lack appropriate focus for what comes next. While many of the points brought up in the paper are important, they have already been stated in many previous papers. The point (I think) is to see how well geodetic and traditional mass balance methods compare over a suitably long period of time (decade). Perhaps focus on the point that analysis over shorter intervals may miss important processes that reveal themselves for longer periods. At the top of page three we first learn where the paper is going. Please state your objectives earlier and reduce introduction by about 50%. A reader should know at about page 1.5 where we are heading.

B) Reorganization required – I appreciate the detailed attention that the authors pay to processes that could make traditional and geodetic methods differ, but the current or-

ganization of these sections comes after key equations used to convert volume change into mass (w.e.) change. You really should present sections (4.1, 4.2 ...) before you present equations (1) and (2). This is especially evident when one reads section 4.2 and then needs to consider whether equation (5) really differs from equation (2) – it doesn't really. This change would make your paper easier to read (certainly more logical).

C) Spatial noise – On page 7 the authors discuss using SDz from stable control area to define spatial variability, but I don't understand how this would yield that information. These control patches serve as so-called 'check points' used in traditional photogrammetry. What would they tell us about spatial variability and how it might affect their results? Not much I'm afraid. What would yield that information, however, is the decorrelation length inherent in their data. The authors have gridded data where they can correct their sample sizes for spatial autocorrelation. You should assess the degree of spatial correlation of your data and reduce number of independent samples accordingly. There are several key papers on this topic, one of them (Rolstad et al., 2009) is cited below.

D) Dimensionless conversion factor K - I have a few problems with the introduction of this variable (K) into the literature. First, this is something that is routinely applied in sequential DEM differencing in many previous studies even though it isn't always stated as such. Second, unless I've missed something K should range between 0-1 yet it is state as ranging between 820-930 (line 267). Third, on lines 386-387 the authors state that their new dimensionless conversion factor K now has units of kg m-3. Many have used this conversion factor in past studies; it's not new, so please let's not re-invent the wheel and muddle the literature with new dimensionless numbers.

E) Clearer discussion needed for explaining discrepancies - One of the major conclusions of this paper is that based on the geodetic balance calculations the authors feel that the years 2002/3, 2005/6, and 2006/7 are biased in the traditional mass balance data. I think they are trying to state that the glacier lost most of its accumulation area

and the bias was caused by having no stakes high up on the glaciers (in this case probing and pits would yield nothing). This point isn't as clear as it need to be in lines 410-444; they need to shorten this section, explicitly implicate the methodological factors that could account for the error and then implicate meteorological factors. As it stands they start with the latter without a clear discussion of the former.

F) Avoid overly bold statements - A minor point, but it is best to avoid absolute statements in papers. The authors suggest that their study is the first to compare annually-resolved geodetic and traditional mass balance records, yet a quick literature search indicates that this isn't correct. For example, Beedle et al, 2014 did this for a shorter period of record and Krimmel (1999) did this for a longer period of time. You should either modify your statements to reflect that your comparison exceeds those of other studies or simply drop statements like this. My preference would be to do the latter.

Specific Comments (editorial and of minor/moderate substance):

A clunky title. I'd suggest. 'Geodetically corrected (or Homogenized) mass balance series of Hintereisferner Glacier, Austria for the period 2001-2011'

First sentence needs to be reworded. It sounds like you obtained 2001-11 mass balance(s) records...

Line 18: Sentence needs revision (grammatically incorrect)

Line 23: Replace 'as a substitute for' with 'superior to'

Line 39: Delete 'and within the snow' since the top of this layer defined glacier surface by definition.

Line 40: Replace 'subtracts' with 'differences'

Line 45: Full stop missing after 'glacier'

Line 50: Beedle et al., (2014) is missing from this list

Line 63-66: Confusing and poorly worded sentences. Please revise.

Line 67: See major comment (F)

Line 86: Add 'an average' after 'with' and strike 'in average'

Line 86: What is a 'totalizing rain gauge' - bulk collector?

Lines 90-94: Tangential to paper's focus (delete).

Lines 100-101: Add 'Annual' at start of sentence, strike 'annual mass balance' and 'have been started' and replace with 'commenced' and strike 'are carried out regularly since then'

Line 129: Strike 'among others' - meaningless in its usage here.

Line 132: 'Further explanation. . .' - Unclear why this statement is here. Reads like an orphaned one.

Line 143: replace 'wrong' with 'incorrect'

Line 148: 'For extrapolating ... ' - This sentence is linked to nothing (a single thread). Not sure why it is here.

Line 154: Replace 'according to the law of error propagation' with 'by error propagation' - There are few physical laws.

Line 180: I had commented in the paper margin 'are density differences treated per elevation band' and hence my suggestion for you to move sections 4.1 and 4.2 before equation (1). See major point (B).

Lines 198-200: See major point (C ).

Lines 212 (and throughout paper): Try not to state things like 'Figure 2 shows. . .'. State trend, observation and refer to figure at end of the sentence. For example, 'Density increases with elevation (Figure 2)'. This allows reader to digest your point and then refer to figure (it also reduces verbiage).

Line 220: Move 'significantly' before 'influence'

Line 228: Add 'absolute' before 'vertical' and strike vertical lines as they are impossible to see in running text.

Line 231: Strike 'very' and avoid this vague qualifier at all costs.

Line 265: See point (D).

Line 271: Replace 'a multi methodical approach was applied incorporating' with 'we incorporated'

Line 276: remove (s) from extrapolations

Line 280-281: How does this standard lapse rate compare to one assessed with station data. Does this help to explain differences in the extreme melt years?

Line 291: Replace '5' with five. Write out all numbers less than 10 unless number has a unit. For example, seven stakes but 3 cm.

Lines 305-212: So how does this approach potentially affect your results? So if you simply ignored effects of crevasses what would results show?

Line 326: Stylistic point, but 'frictional dissipation' I believe is the more precise term. Line 348: Write out 'E.g. - Never start a sentence with this.

Line 353: If it's a small term at the annual scale, it's small and within error at decadal scale. It can't be significant for one but not the other. Suggest dropping last clause in sentence.

Line 357: 'The 2001 to 2011 one step...' - Not sure what sentence is trying to state.

Line 358-359: See earlier comment about 'Table and Figure shows...'

Line 363: What does 'respectively' refer to? The penultimate sentence? Revise.

Line 370: Sometimes last word before equation has a colon sometimes not, be consistent with journal standards.

Line 371: How does $\delta$ change if you incorporate effective degrees of freedom in the geodetic estimate of uncertainty (i.e. correct for spatial autocorrelation)?

Line 376: Replace 'coherent' with 'similar'

Line 378: How is bias defined in this paper? Should be formally defined.

Line 383: Did you really explore the parameter space? This phrase is typically used with Monte Carlo sampling or Latin Hypercube sampling. Did you do that?

Line 388: Why does K now have units? You told us earlier that it was dimensionless. . .

Line 400: If you used results that weren't smoothed (removal of crevasses) how does this affect your results?

Lines 410-440: I found this portion extremely difficult to follow for reasons outlined in major point (E).

Line 448: See point (F). Several papers out there that do this. Your paper, however, does this for the longest series, and will be well received. But please don't oversell its novelty.

Line 452: Change sentence to, 'It neither include(s) a through. . . nor '

Line 453: Remove 'ed' from showed.

LIne 456: Change 'a snow cover' to 'snow'

References: I did not check these for typos, but suggest you add the ones in this review to the list.

Figures:

Fig. 2 - A legend added to this figure would help reader. It would be nice in the figure caption to state level of uncertainty (68, 95%).

Fig. 3 - What are units of Intensity (DN?). $kg\ m^{-3}$.

Fig. 4. - You don't deal with dynamics (flux divergence) so it is not appropriate to plot these data as 'Mass balance [m w.e.]' as a function of elevation. $\frac{dh}{dt}$ and $\b{dot}$ are not equal due to dynamics. This plot must be redrafted showing 'Elevation change [m w.e.] and not 'Mass balance'.

Fig. 6. Remove titles from figures and simply use 'a)' and 'b)' . Avoid excessive qualifiers. Change 'The extraordinary mass' to 'Mass'

Fig. 9. Is this really the best way to show these data? Why not simply remove the figure and tell reader in text if homogenized hintereisferner series correlates more strongly (or use of other statistic than Pearson) with nearby series.

Tables:

There are a lot of them and not sure if they are all needed. Any individual wanting your data would request them, no? Alternatively you could deposit them with the WGMS or other agency (or include as electronic supplementary data). They take up a lot of journal space and some repeat what figures show.

Table 5. Replace 'cum' with 'Sum'.

References:

Beedle, Matthew J., Brian Menounos, and Roger Wheate. 2014. "An Evaluation of Mass-Balance Methods Applied to Castle Creek Glacier, British Columbia, Canada." Journal of Glaciology 60 (220): 262–76.

Krimmel, Robert M. 1999. "Analysis of Difference Between Direct and Geodetic Mass Balance Measurements at South Cascade Glacier, Washington." Geografiska Annaler: Series A, Physical Geography 81 (4). Blackwell Publishers Ltd: 653–58.

Rolstad, C., T. Haug, and B. Denby. 2009. "Spatially Integrated Geodetic Glacier Mass Balance and Its Uncertainty Based on Geostatistical Analysis: Application to the."

Journal of Glaciology 55 (192). http://www.igsoc.org:8080/journal/55/192/j08j136.pdf.

---

## Author Comment (AC2) · 17 Nov 2017

**A reanalysis of one decade of the mass balance series on Hintereisferner, Ötztal Alps, Austria: a detailed view into annual geodetic and glaciological observations**

Author's reply to referee comments

Christoph Klug and Co-authors
November 17, 2017

**Introductory Remarks**

We thank the anonymous Referee for reviewing our manuscript and the thoughtful and constructive comments. Our response letter is structured as follows. Section 1 provides detailed answers to general concerns raised by the referee, whereas section 2 offers a point by point response to the specific remarks.

To facilitate readability, the referee's comments are given in *grey italics* while our responses are in blue regular font.

**1.    Reply to general referee-comments #2 by Anonymous Referee**

*Klug and co-authors compare annually-resolved geodetic (airborne laser altimetry) and traditional mass balance records of Hintereisferner Glacier over the period 2001-2010. They find that for most years these two methods estimate similar mass change for the glaciers (within uncertainties). They note discrepancy between the methods for three years and attribute these differences to errors in the traditional mass balance data.*

*I found this paper (and their experimental design) to be well thought out and mostly well described. It should be published. Globally, the number of traditional mass balance records relative to the number of existing glaciers is vanishingly small. We thus require alternative approaches that could complement conventional methods of measuring surface mass balance. Klug and co-authors make a convincing case that estimating mass balance using geodetic techniques is appropriate and in some cases can identify errors in the traditional series. This is especially important when one uses these series for calibration of models (hydrologic and mass balance).*

*I would recommend that the paper be considered for publication, but only after the authors make a number of substantive changes. Most of these suggestions are minor, but several will require some thought and future analysis. Below, I expand on the major points the authors should address. In several places I found the writing to be muddled and fraught with logic gaps and grammatical/typographical errors that might arise if English was not the first language of the senior author. I would encourage all authors spend time going through the revised manuscript to ensure its presentation is up to the standards required for the journal.*

**GENERAL COMMENTS**

*A) Streamline introduction - I found the introduction of the paper to be too long and lack appropriate focus for what comes next. While many of the points brought up in the paper are important, they have already been stated in many previous papers. The point (I think) is to see how well geodetic and traditional mass balance methods compare over a suitably long period of time (decade). Perhaps focus on the point that analysis over shorter intervals may miss important processes that reveal themselves for longer periods. At the top of page three we first learn where the paper is going. Please state your objectives earlier and reduce introduction by about 50%. A reader should know at about page 1.5 where we are heading.*

We widely agree with the argumentation of the referee. The revised introduction is significantly shorter than the original one. We changed the introduction section with the aim of clearly showing the background, motivation and starting point of the paper. This was reached by omitting passages containing information which is common knowledge within the community or which is not relevant for the reader in this part of the paper. Thereby the main objectives of the study are presented earlier in the paper. We also tried to sharpen the motivation of the paper by pointing out the research focus more clearly.

*B) Reorganization required – I appreciate the detailed attention that the authors pay to processes that could make traditional and geodetic methods differ, but the current organization of these sections comes after key equations used to convert volume change into mass (w.e.) change. You really should present sections (4.1, 4.2 . . .) before you present equations (1) and (2). This is especially evident when one reads section 4.2 and then needs to consider whether equation (5) really differs from equation (2) – it doesn't really. This change would make your paper easier to read (certainly more logical).*

Indeed the structure of the paper was a point of long discussions between the authors. We also agree that the organization of the chapters still provides some challenges for the reader. Nevertheless, after detailed discussions and thorough evaluation of the reviewer's suggestion, we prefer to keep the structure of the manuscript unchanged due to the following reason
In section 4 the inherent differences between two methods are addressed in detail. It is impossible to start such a discussion before the two methods are introduced.

*C) Spatial noise – On page 7 the authors discuss using SDz from stable control area to define spatial variability, but I don't understand how this would yield that information. These control patches serve as so-called 'check points' used in traditional photogrammetry. What would they tell us about spatial variability and how it might affect their results? Not much I'm afraid. What would yield that information, however, is the decorrelation length inherent in their data. The authors have gridded data where they can correct their sample sizes for*

*spatial autocorrelation. You should assess the degree of spatial correlation of your data and reduce number of independent samples accordingly. There are several key papers on this topic, one of them (Rolstad et al., 2009) is cited below.*

We did not intend to show the spatial variability of the DTM errors, but to give a measure on the overall DTM accuracy affecting the geodetic mass balances. Since the errors are quite low and do not show large spatial variation within the DTM, this was deemed a comprehensible approach. In our case of ALS (> 1 point per m2) it can be assumed that the number of independent items is about the number of glacier pixels (cf. Joerg et al. 2012). However, since both reviewers criticized this, we changed the way in which we calculate the random error of the used DTM. This also leads to a much lower random error in the DTM.
Although we are aware that there is also the implicit assumption that the DTM uncertainty over stable terrain is representative for the DTM uncertainty over the glacier (cf. Rolstad et al. 2009, J. Glaciol.), we did not correct our sample sizes for spatial autocorrelation, but added a better discussion of this issue to the revised manuscript.

***D) Dimensionless conversion factor K** - I have a few problems with the introduction of this variable (K) into the literature. First, this is something that is routinely applied in sequential DEM differencing in many previous studies even though it isn't always stated as such. Second, unless I've missed something K should range between 0-1 yet it is state as ranging between 820-930 (line 267). Third, on lines 386-387 the authors state that their new dimensionless conversion factor K now has units of kg m-3. Many have used this conversion factor in past studies; it's not new, so please let's not re-invent the wheel and muddle the literature with new dimensionless numbers.*

There seems to be a misunderstanding. We did not want to introduce this factor to the scientific community, only to our paper. By giving units of kg m-3 we tried to make the factor more comparable.
We are well aware of the fact that this is not a novel approach and therefore we added some respective references to the revised manuscript to clarify this. Furthermore, we adapted the manuscript in order to assure a consistent use of the dimensionless K throughout the paper.

***E) Clearer discussion needed for explaining discrepancies** - One of the major conclusions of this paper is that based on the geodetic balance calculations the authors feel that the years 2002/3, 2005/6, and 2006/7 are biased in the traditional mass balance data. I think they are trying to state that the glacier lost most of its accumulation area and the bias was caused by having no stakes high up on the glaciers (in this case probing and pits would yield nothing). This point isn't as clear as it need to be in lines 410-444; they need to shorten this section, explicitly implicate the methodological factors that could account for the error and then*

*implicate meteorological factors. As it stands they start with the latter without a clear discussion of the former.*

We agree with the referee. In the revised manuscript we attribute the differences between the mass balance methods more clearly to an insufficient measurement set-up and missing observations in the former accumulation area. Furthermore we shortened and restructured the section which results in a more logic streamflow of discussion which is easier to follow for the reader.

*F) Avoid overly bold statements - A minor point, but it is best to avoid absolute statements in papers. The authors suggest that their study is the first to compare annually resolved geodetic and traditional mass balance records, yet a quick literature search indicates that this isn't correct. For example, Beedle et al, 2014 did this for a shorter period of record and Krimmel (1999) did this for a longer period of time. You should either modify your statements to reflect that your comparison exceeds those of other studies or simply drop statements like this. My preference would be to do the latter.*

Beedle et al., 2014 applied photogrammetry based geodetic mass balances for three years while Krimmel et al., 1999 used digitized maps and photogrammetry over multi-year periods. To our knowledge our study presents the first approach based on annual high resolution ALS DEMs over the period of one decade.. However we omitted the word"first" and also dropped down similar statements throughout the paper. Furthermore we explained the innovative aspect of our study more clearly in the revised manuscript.

**2.    Reply to specific referee-comments #2 by an Anonymous Referee**

**SPECIFIC COMMENTS** (editorial and of minor/moderate substance)

**Title:** *A clunky title. I'd suggest. 'Geodetically corrected (or Homogenized) mass balance series of Hintereisferner Glacier, Austria for the period 2001-2011'*
In the revised manuscript the title has been changed

**Line 14:** *First sentence needs to be reworded. It sounds like you obtained 2001-11 mass balance(s) records.*
Done.

**Line 18:** Sentence needs revision (grammatically incorrect)
Done

**Line 23:** Replace 'as a substitute for' with 'superior to'
Done.

**Line 39:** Delete 'and within the snow' since the top of this layer defined glacier surface by definition.
Done.

**Line 40:** Replace 'subtracts' with 'differences'
Done.

**Line 45:** Full stop missing after 'glacier'
Done.

**Line 50:** *Beedle et al., (2014) is missing from this list*
We added this reference to the revised manuscript.

**Line 63-66:** Confusing and poorly worded sentences. Please revise.
Done.

**Line 67:** See major comment (F)
Done.

**Line 86:** Add 'an average' after 'with' and strike 'in average'
Done.

**Line 86:** What is a 'totalizing rain gauge' - bulk collector?

We use the term *totalizer rain gauge* as it is defined by the meteorology glossary from the American Meteorological Society: http://glossary.ametsoc.org/wiki/Totalizer_rain_gauge

**Lines 90-94:** Tangential to paper's focus (delete).

Done.

**Lines 100-101:** Add 'Annual' at start of sentence, strike 'annual mass balance' and 'have been started' and replace with 'commenced' and strike 'are carried out regularly since then'

Done.

**Line 129:** Strike 'among others' - meaningless in its usage here.

Done.

**Line 132:** 'Further explanation. - Unclear why this statement is here. Reads like an orphaned one.

The respective sentence was deleted.

**Line 143:** replace 'wrong' with 'incorrect'

Done.

**Line 148:** 'For extrapolating ... ' - This sentence is linked to nothing (a single thread). Not sure why it is here.

The respective sentence was deleted.

**Line 154:** Replace 'according to the law of error propagation' with 'by error propagation' - There are few physical laws.

Done.

**Line 180:** I had commented in the paper margin 'are density differences treated per elevation band' and hence my suggestion for you to move sections 4.1 and 4.2 before equation (1). See major point (B).

We kindly refer to our detailed response in  part (B) of the general comments.

**Lines 198-200:** See major point (C).

We kindly refer to our detailed response in part (C) of the general comments.

**Lines 212 (and throughout paper):** Try not to state things like 'Figure 2 shows. . .'. State trend, observation and refer to figure at end of the sentence. For example, 'Density increases

with elevation (Figure 2)'. This allows reader to digest your point and then refer to figure (it also reduces verbiage).

We  avoid such  throughout the revised manuscript.

**Line 220:** Move 'significantly' before 'influence'

Done.

**Line 228:** Add 'absolute' before 'vertical' and strike vertical lines as they are impossible to see in running text.

Done.

**Line 231:** Strike 'very' and avoid this vague qualifier at all costs.

Done.

**Line 265:** See point (D).

We refer to our response in part (D) of the general comments.

**Line 271:** Replace 'a multi methodical approach was applied incorporating' with 'we incorporated'

Done.

**Line 276:** remove (s) from extrapolations

Done.

**Line 280-281:** How does this standard lapse rate compare to one assessed with station data. Does this help to explain differences in the extreme melt years?

In the case of significant snow fall we assume the atmosphere to be saturated. Furthermore the analysis of station data shows that the use of the moist adiabatic gradient in such cases is quite a reasonable assumption. However, the large differences between the two mass balance methods are not sensitive to the choice of this lapse rates  which  hence do not significantly contribute to explaining those differences. We added a statement discussing this issue to the revised manuscript.

**Line 291:** Replace '5' with five. Write out all numbers less than 10 unless number has a unit. For example, seven stakes but 3 cm.

Done.

**Lines 305-212:** So how does this approach potentially affect your results? So if you simply ignored effects of crevasses what would results show?

This approach does not affect our results significantly; it is only included to show that the effects of crevasses are negligible to our results. We expanded section 4.4 to explicitly highlight this.

**Line 326:** Stylistic point, but 'frictional dissipation' I believe is the more precise term.
Done.

**Line 348:** Write out 'E.g. - Never start a sentence with this.
Done.

**Line 353:** If it's a small term at the annual scale, it's small and within error at decadal scale. It can't be significant for one but not the other. Suggest dropping last clause in sentence.
Done.

**Line 357:** 'The 2001 to 2011 one step. . .' - Not sure what sentence is trying to state.
The sentence was revised for clarity.

**Line 358-359:** See earlier comment about 'Table and Figure shows. . .'
Done.

**Line 363:** What does 'respectively' refer to? The penultimate sentence? Revise.
The sentence was chanced for clarity.

**Line 370:** Sometimes last word before equation has a colon sometimes not, be consistent with journal standards.
Done.

**Line 371:** How does $\delta$ change if you incorporate effective degrees of freedom in the geodetic estimate of uncertainty (i.e. correct for spatial autocorrelation)?
We kindly refer to our detailed response in the general comments (part C).

**Line 376:** Replace 'coherent' with 'similar'
Done.

**Line 378:** How is bias defined in this paper? Should be formally defined.
Done.

**Line 383:** Did you really explore the parameter space? This phrase is typically used with Monte Carlo sampling or Latin Hypercube sampling. Did you do that?

No we just explored the minimum and maximum random uncertainties. We added a clarifying statement to the revised manuscript.

**Line 388:** Why does K now have units? You told us earlier that it was dimensionless. . .

We only introduced the units to make it more comparable for the reader. This has been amended following the suggestions by the reviewer.

**Line 400:** If you used results that weren't smoothed (removal of crevasses) how does this affect your results?

As already shown in table 5, and section 4.4, the effect on the results is negligible. We added a reference to table 5 and section 4.4. to the respective paragraph.

**Lines 410-440:** I found this portion extremely difficult to follow for reasons outlined in major point (E).

We refer to our response in part (E) of the general comments.

**Line 448:** See point (F). Several papers out there that do this. Your paper, however, does this for the longest series, and will be well received. But please don't oversell its novelty.

We refer to our response in part (F) of the general comments.

**Line 452:** Change sentence to, 'It neither include(s) a through. . . nor '

Done.

**Line 453:** Remove 'ed' from showed.

Done.

**Line 456:** Change 'a snow cover' to 'snow'

Done.

*References:*

I did not check these for typos, but suggest you add the ones in this review to the list.

Done.

*Figures:*

**Fig. 2** - A legend added to this figure would help reader. It would be nice in the figure caption to state level of uncertainty (68, 95%).

Done.

**Fig. 3** - What are units of Intensity (DN?). kg m$^{-3}$?
Intensity has no units. The used intensity rasters only show the backscattered energy stored in 8 bit or 256 grey values.

**Fig. 4.** - You don't deal with dynamics (flux divergence) so it is not appropriate to plot these data as 'Mass balance [m w.e.]' as a function of elevation. $\frac{dh}{dt}$ and $\dot{b}$ are not equal due to dynamics. This plot must be redrafted showing 'Elevation change [m w.e.] and not 'Mass balance'.

It is true that we do not explicitly resolve ice dynamics. Nevertheless, the change in surface elevation can be used to calculate a mass balance which is the result of accumulation/ablation processes (surface, internal, basal) and ice flux divergence. In principle it does not matter if this is done for a point/column, an elevation band or the whole glacier (e.g. Cogley et al., 2011, page 5). However, we think that comparisons of geodetic and direct balances are only problematic if not done on the glacier wide scale. Hence, we did not redraft the plot but added a statement discussing the effect of ice dynamics on the local mass balance and the implications on method-comparisons on scales others than "glacier-wide".

**Fig. 6.** Remove titles from figures and simply use 'a)' and 'b)' . Avoid excessive qualifiers. Change 'The extraordinary mass' to 'Mass'
Done.

**Fig. 9.** Is this really the best way to show these data? Why not simply remove the figure and tell reader in text if homogenized Hintereisferner series correlates more strongly (or use of other statistic than Pearson) with nearby series.
This figure supports what we discuss in lines 436 to 448 of the original manuscript and was hence kept in the paper. However, we added a clearer statement on correlation with other glaciers to the revised manuscript.

**Tables:**
There are a lot of them and not sure if they are all needed. Any individual wanting your data would request them, no? Alternatively you could deposit them with the WGMS or other agency (or include as electronic supplementary data). They take up a lot of journal space and some repeat what figures show.

While it is true that a lot of tables are included in the manuscript, we do think that they add important information. However, we decided to move table 4 from the manuscript to the supplementary data. If required, we will move more of the tables to the supplements, but we would prefer to keep them within the paper.

**Table 5.** Replace 'cum' with 'Sum'.
Done.

**References:**

Beedle, Matthew J., Brian Menounos, and Roger Wheate. 2014. "An Evaluation of Mass-Balance Methods Applied to Castle Creek Glacier, British Columbia, Canada." Journal of Glaciology 60 (220): 262–76.

Krimmel, Robert M. 1999. "Analysis of Difference between Direct and Geodetic Mass Balance Measurements at South Cascade Glacier, Washington." Geografiska Annaler: Series A, Physical Geography 81 (4). Blackwell Publishers Ltd: 653–58.

Rolstad, C., T. Haug, and B. Denby. 2009. "Spatially Integrated Geodetic Glacier Mass Balance and Its Uncertainty Based on Geostatistical Analysis: Application to the." Journal of Glaciology 55 (192). http://www.igsoc.org:8080/journal/55/192/j08j136.pdf.

Those references have been added.

---

## Author Comment (AC1)

**A reanalysis of one decade of the mass balance series on Hintereisferner, Ötztal Alps, Austria: a detailed view into annual geodetic and glaciological observations**

Author's reply to referee comments#1

Christoph Klug and Co-authors
November 17, 2017

**Introductory Remarks**

We thank Michael Zemp for reviewing our manuscript and the thoughtful and constructive comments. Our response letter is structured as follows. Section 1 provides detailed answers to general concerns raised by the referee, whereas section 2 offers a point by point response to the specific remarks.

To facilitate readability, the referee's comments are given in *grey italics* while our responses are in blue regular font.

**1. Reply to general referee-comments #1 by M. Zemp**

*Christoph Klug and colleagues present a detailed reanalysis of annual glaciological and annual geodetic balances at Hintereisferner, Austria, obtained between 2001 and 2011. This study puts an airborne laser scanner (ALS) dataset with exceptional spatial and temporal resolution over an entire decade at its full value. The comparison of these geodetic results with the glaciological balances from an extensive in-situ network have been long overdue but are now carried out in a very thorough way and including an error assessment according to best international practises. Hence, I can recommend the paper for publication in The Cryosphere after consideration of the following two substantial points and a list of suggestions for minor corrections and clarifications:*

**GENERAL COMMENTS**

*1.* **_DTM-related random uncertainty of geodetic balances:_**

*The authors use the standard deviation of the DTM-differencing over selected stable terrain as random uncertainty for the geodetic balance (cf. equation 3, lines 196-207). I do not agree with this approach because it assigns a local DTM error to a zonal glacier change value. The standard deviation of the elevation differences on stable terrain indicates the uncertainty of the DTM differences for individual pixels. Instead, I propose to use the standard error, defined as the standard deviation divided by the square root of the number of independent items of information in the sample (cf. Zemp et al. 2013, The Cryosphere, Section 2.3). In the*

*present case of ALS (> 1 point per m2) it can probably be assumed that the number of independent items is about the number of glacier pixels (cf. Joerg et al. 2012, RSE). Note that there is also the implicit assumption that the DTM uncertainty over stable terrain is representative for the DTM uncertainty over the glacier (cf. Rolstad et al. 2009, J. Glaciol.). Maybe that needs just to be mentioned somewhere in the paper.*

We followed the suggestions of the reviewer for this point. We therefore calculated the standard deviation and divided it by the square root or the number of independent items. This of course leads to a significantly lower standard error. We of course assume comparable DTM uncertainties over the whole DTM, which therefore do not differ between stable areas and glaciated terrain. We also stated this more clearly in section 3.2.

**2.        *Geodetic method as substitution for the glaciological method:**

*The authors conclude that the geodetic method (i) "can represent a valuable possibility to overcome shortcoming in the glaciological measurements even on an annual scale" (Lines 469-470) or (ii) "even as a substitute for the glaciological method". I can only partly support these conclusions for three reasons:*
*(1) the geodetic and the glaciological methods are rather complementary in nature (than to substitute each other): the strength of the glaciological method is to capture the spatial and temporal variability of the glacier surface balance even with only a small sample of observation points but it is sensitive to systematic errors which accumulate linearly with the number of seasonal or annual measurements. The geodetic balance is able to cover the entire glacier but requires a density conversion, which becomes more challenging over short time periods because of meteorological influences on the elevation change.*

We agree with the reviewer and changed the manuscript accordingly, especially regarding the wording and the complementary nature of the two methods. We tried to elaborate more comprehensively why a reanalysis based on geodetic data is needed for HEFs glaciological mass balance record. We also agree that the strength of the glaciological method is the ability to capture spatial and temporal (year to year) variability of surface mass balance and to extract the part of mass change which is a consequence of meteorological forcing. However, this is only given if the analyses follow a certain quality standard. In terms of unexplainable differences between the methods, a thorough uncertainty assessment has to be conducted in order to indicate that available glaciological balances are questionable and geodetic data can help in improving shortcoming in the glaciological measurements.

*(2) the nature of uncertainties: typically, ten years of data are required for the detectable difference to become lower than the annual random "noise" of the glaciological balance (cf. Zemp et al. 2013, The Cryosphere). A validation at annual time intervals might actually miss a bias.*

This is correct and the reason we included the comparison for the ten-year period. The annual uncertainties are not suitable for bias detection, but the entire period of our investigation is. We have highlighted and discussed this issue in the revised manuscript. The annual uncertainties are only used to identify years which differ significantly between the two methods.

*(3) cost-benefit considerations: the costs of the geodetic method are one to two orders of magnitudes higher than the costs of the glaciological method.*

Regarding the cost-benefits, of course those depend on the individual investigated glaciers and the methodological set-up. In the case of Hintereisferner the cost of an ALS-campaign is about 50 to 100% higher than the budget for the labour intense direct measurements which (if done properly) require highly qualified stuff. However, this point is discussed more critically in the revised manuscript.

*I suggest adding a short section that discusses these issues and rewording the corresponding conclusions.*

Done.

**2.    Reply to specific referee-comments #1 by M. Zemp**

**MINOR CORRECTIONS AND CLARIFICATIONS**

**Page3, Line67:**

*"first use of annual geodetic records": At South Cascade Glacier, annual results from both geodetic and glaciological methods have been analysed by Krimmel (1999). Robert M. Krimmel (1999) Analysis of difference between direct and geodetic mass balance measurements at South Cascade Glacier, Washington, Geografiska Annaler: Series A, Physical Geography, 81:4, 653-658.*

The authors meant the first use of annual ALS data. However "first" was cleared.

**P4, L119:**

*"Results are submitted to the WGMS. . .": you could add a reference to WGMS (2017, and earlier reports): WGMS 2017. Global Glacier Change Bulletin No.2 (2014–2015). Zemp, M., Nussbaumer, S. U., Gärtner- Roer, I., Huber, J., Machguth, H., Paul, F., and Hoelzle, M. (eds.), ICSU(WDS)/IUGG(IACS)/UNEP/UNESCO/WMO, World Glacier Monitoring Service, Zurich, Switzerland.*

Done. Reference was implemented.

**P6, L178-184:**

*Equation 2: the geodetic balance is usually calculated using the average glacier area of the two surveys (cf. Zemp et al. 2013, The Cryosphere, Eq. (5) and (6)). At annual time steps, this might not make a big difference, but for the decadal period with a surface area reduction of 15% it does become relevant.*

We agree. Method of calculation for decadal period was adapted to using the time averaged area mean $(S_{t1}+S_{t2})/2$, as recommended in Zemp et al., 2013.

**P6, L188 & Fig 1:**

*Stable areas: I fully support the decision to complement the down-valley soccer field with stable areas near the glacier. Please add a short comment about the selection criteria for the stable areas A-E.*

Visual inspection and expert knowledge of the terrain (Sailer et al., 2012, 2013; Bollmann et al., 2011). Comment was added.

**P8/9, L240-267:**

*Density conversion: the density conversion factor depends on changes in the three-dimensional firn body and is a function of (i) the additional snow layer incl. related densification and metamorphosis, (ii) firn compaction and metamorphosis, and (iii) sub/emergence velocity. From the text, I cannot fully comprehend how these factors are covered (or not) by the author's approach combining differential DTMs, surface classifications, and density assumptions. Please clarify and discuss the opportunities and limitations of the used approached.*

The conducted density conversion consists of three steps within our approach. First, the dDTM was calculated. In a next step, the glacier surface was classified into two classes (firn and ice) by using the intensity images of the ALS campaign, resulting in surface grids for each year. By subtracting the classified intensity rasters and reclassifying the resulting new surface raster, we incorporated the changing extent of the perennial firn zones in a third step. This should answer point (ii) raised by the reviewer. However, we are aware that firn compaction and metamorphosis are not covered by this approach.

Point (iii) could not really be considered using the available data, which is why we already mentioned in the introduction that we will not incorporate glacier flow dynamics in the presented analysis.

Regarding point (i), the snow layer was incorporated by combining a maximum snow height at the time of measurement with in-situ measured snow-densities, to redistribute the mass according to the snow layer to the glacier surface. Nevertheless, we are aware that this type of spatial distributed density conversion is rather a best guess than a three-dimensional modelling of the firn body.

In the revised manuscript we tried to clarify and discuss our way of density conversion in a more comprehensive way and added a workflow chart (see Figure 1), which helps to better follow the steps within our analysis.

[Figure]

Figure 1: Workflow chart.

**P9, L266-267 & Table 5:**

*Density conversion factor and related uncertainties: for a non-expert it is hard to follow how the density conversion factor and corresponding random uncertainties (together with the annual balance) relate to K.sigma and K.epsilon in Table 5. Adding a corresponding equation in Section 4.2 might help.*

We agree with the referee and added a corresponding equation.

**P9, L271:**

*"stratigraphic year": I think this should be "end of the hydrological year" or "fixed date system" (cf. P9, L275, "30th September").*

Changed accordingly to "end of hydrological" year

**P10, L285-287:**

*"elevation dependent mean ablation gradient": do you use the same gradient for the ablation and the accumulation zone? Please clarify.*

The ablation gradient is derived from in situ stake readings (L 278). The gradient is applied to correct the survey date difference between geodetic and glaciological survey in the years 2003 and 2008 (L 285-287). Table 4 shows that in 2003 (survey date correction 4 days) and 2008

(21 days) snow cover was present, varying with altitude. Thus, the ablation at each individual stake is a bulk information consisting of snow and ice ablation. We average the stake ablation per 100 m elevation zone and hence derive the elevation dependent ablation gradient along the elevation range of the glacier.

To avoid confusion with the word *mean*, we deleted *mean* (L 287) as it should refer to the mean of the elevation zone and not to a mean value all over the glacier.

**P11, L323-324:**

*for comparability, convert the values by Thibert et al. (2008) to annual change rates.*
Done.

**P15, L448-451:**

*"were the first and so far only": consider rewording in view of earlier studies at South Cascade by Krimmel (1999, Geogr. Ann.).*
Done.

**Text, Figs & Tabs, "altitude" versus "elevation"***:*

*In most cases, you could replace "altitude" by "elevation" (cf. McVicar, T. R., & Körner, C. (2013). On the use of elevation, altitude, and height in the ecological and climatological literature. Oecologia, 171(2), 335-337.)*
Done.

**P24, Fig. 1:**

*For clarification, you could write in the figure caption: "Note that in 2003, no accumulation measurements COULD have been carried out DUE TO THE STRONGLY REDUCED ACCUMULATION ZONE. HENCE, only ablation stakes were available."*
Done.

**P25&30, Fig. 2 & 7:**

*the two figures are redundant to a certain degree. On the other side, it is not fully clear, which differences and uncertainties are included. Please at least clarify in captions. In addition, you could consider merging Fig 2 & 7, showing bias corrections for both glaciological and geodetic results. Instead, you could remove the cumulative curves (=> shown in Fig. 8).*

We agree with the referee and will merge the two figures in the revised manuscript. Furthermore, we clarified in captions which uncertainties are included and removed the cumulative curves.

**P26. Fig. 3:**

*I would add a bar showing the intensity range (values) to the legend of the left image. In the legend of the right one, I would replace "perennial firn" by "snow and firn".*
Done

**P27, Fig. 4:**

*In the caption, please clarify what you mean with "Corrected". It might be sufficient adding a reference to the corresponding section in the paper. I would add the terms glaciological and geodetic to the label of the x-axis in the left and right figure, respectively. In addition, please add a note on the effect of the sub/emergence velocity.*
Done

**P28, Fig. 5:**

*you could add the data point(s) for the full period (glaciol.cum versus geod.cum, glaciol.cum versus geod.01/11).*
Done

**P29, Fig. 6:**

*Please add a note on the effect of the sub/emergence velocity.*
The following note on the influence of glacier dynamics was added:
Due to ice dynamics, an underestimation of the geodetic versus the glaciological mass balance is expected in the accumulation area (and vice versa in the ablation area). However, the surface height change due to the flux divergence is at least one order of magnitude smaller than the values presented here.

**P31, Fig. 8:**

*typically, one would calibrate the glaciological with the geodetic over the decadal period (i.e. 2001-11). Hence, it might be good to show that result here too.*
Done

**P34, Tab. 2:**

*you could add a column for the two dDTM of the full period, i.e. 01/11.*
Done

**P35, Tab 3:**

*please explain why the density given in the caption (900 kg m-3) differs with the one mentioned in the text (850 kg m-3, cf. P8, L249)*
It's a typo and was clarified!

**P36, Tab 4:**

*in the caption, there are some problems with the symbol for average SC. What is the "mean acc. area"? Do you refer to the end-of-summer accumulation area?*

Symbol problems have been revised. Mean accumulation area is the classified firn area ($A_F$). To avoid ambiguity it was changed.

**P37, Tab 5:**

*I would expect the annual uncertainties for the density conversion (sigma K) to be larger than for the (zonal) ones for the ALS-DTM (sigma DTM). See also my comments above (substantial point (a) and comment related to density conversion, P9, L266 267).*

Since we changed the calculation of the errors for the ALS-DTM (sigma DTM), those are now lower than in the originally submitted manuscript.

**P38, Tab 6, caption:**

*consider rewording "improved balance" into "bias-corrected balances"; consider rewording "statistical significance" by "reduced discrepancy". Use the same symbol for the common variance in caption (now wrongly epsilon.comvar) and table (=> sigma.comvar).*

Done

---

## Author Response (AR1)

Dear Dr. Radić,

to enhance the quality of our paper we have, in the end, resolved some of the reviewers criticisms differently than previously indicated in the initial response to the reviews. Please find our updated responses to the comments of the two referees, as well as the indication of the respective changes applied to the revised manuscript below.

While Section 1 of the present document provides detailed answers to the most important concerns raised by the two referees, Sections 2 and 3 offer a point by point response to their specific remarks. The referee's comments are given in *grey italics* while our responses are in blue regular font.

In reaction to the referee comments the revised version of our manuscript includes a number of changes. The most substantial ones are the following:

- We have significantly shortened the introduction and pointed out the aim of the paper more explicitly as suggested by referee II.

- As demanded by referee II we re-structured the manuscript. This was achieved by moving parts of the former introduction to Section 3 and by revising and shortening Sections 4 and 5 of the original manuscript. Furthermore we split the original Section 5 into two sections (5 Results, 6 Discussion). However we did not follow the suggestions of the referee to present Section 4 prior to Equations 1 and 2 (Sect.3) since in our opinion methods have to be introduced in general (Sect. 3) before they are discussed in detail (Sect.4).

- We omitted the misleading density conversion factor and changed Equation 5 which now explains how we calculated our conversion density $\overline{\rho}$ for individual years.

- We recalculated uncertainties related to DEMs as suggested by the referees and changed all concerned numbers and figures in the revised manuscript.

- We changed the symbology of mass balance and uncertainty terms to be in line with Cogley et al. (2011) and Zemp et al. (2013).

- We reduced the number of figures and tables and revised the remaining ones including their labels and captions.

The last part of this document contains a marked-up version of the revised manuscript indicating all the changes applied.

Thank you for your consideration of our revised manuscript for publication in *The Cryosphere*.

Best regards,
Christoph Klug & Stephan P. Galos on behalf of the Co-authors

**1 Author's response to most important referee concerns**

**1.1 Major comments by referee I (M. Zemp)**

**1.1.1 DTM-related uncertainties of geodetic mass balance**

*DTM-related random uncertainty of geodetic balances: The authors use the standard deviation of the DTM-differencing over selected stable terrain as random uncertainty for the geodetic balance (cf. equation 3, lines 196-207). I do not agree with this approach because it assigns a local DTM error to a zonal glacier change value. The standard deviation of the elevation differences on stable terrain indicates the uncertainty of the DTM differences for individual pixels. Instead, I propose to use the standard error, defined as the standard deviation divided by the square root of the number of independent items of information in the sample (cf. Zemp et al., 2013, The Cryosphere, Section 2.3). In the present case of ALS (> 1 point per m2) it can probably be assumed that the number of independent items is about the number of glacier pixels (cf. Joerg et al., 2012, RSE). Note that there is also the implicit assumption that the DTM uncertainty over stable terrain is representative for the DTM uncertainty over the glacier (cf. Rolstad et al., 2009, J. Glaciol.). Maybe that needs just to be mentioned somewhere in the paper.*

In the revised manuscript we followed the referee's suggestion and recalculated the uncertainties related to DTMs according to Zemp et al. (2013). We therefore calculated the standard deviation and divided it by the square root of the number of grid cells. This of course leads to a significantly lower uncertainty. All numbers of concern were adjusted.

Indeed we assume comparable DTM-uncertainties over the whole DTM, which should consequently not differ (significantly) between stable areas and glaciated terrain. We also stated this more clearly in section 3.2 of the revised manuscript.

**1.1.2 Geodetic method as substitution for the glaciological method**

*Geodetic method as substitution for the glaciological method: The authors conclude that the geodetic method (i) "can represent a valuable possibility to overcome shortcoming in the glaciological measurements even on an annual scale" (Lines 469-470) or (ii) "even as a substitute for the glaciological method". I can only partly support these conclusions for three reasons: (1) the geodetic and the glaciological methods are rather complementary in nature (than to substitute each other): the strength of the glaciological method is to capture the spatial and temporal variability of the glacier surface balance even with only a small sample of observation points but it is sensitive to systematic errors which accumulate linearly with the number of seasonal or annual measurements. The geodetic balance is able to cover the entire glacier but requires a density conversion, which becomes more challenging over short time periods because of meteorological influences on the elevation change. (2) the nature of uncertainties: typically, ten years of data are required for the detectable difference to become lower than the annual random "noise" of the glaciological balance (cf. Zemp et al., 2013, The Cryosphere). A validation at annual time intervals might actually miss a bias. (3) cost-benefit considerations: the costs of the geodetic method are one to two orders of magnitudes higher than the costs of the glaciological method. I suggest adding a short section that discusses these issues and rewording the corresponding conclusions.*

We agree with the reviewer and changed the manuscript accordingly, especially regarding the

wording and the complementary nature of the two methods. We removed statements about a possible replacement of the glaciological by the geodetic method. We tried to elaborate more comprehensively why a reanalysis based on geodetic data is needed for HEFs glaciological mass balance record and what the benefit of such a reanalysis is.

We also agree that the strength of the glaciological method is the ability to capture spatial and temporal (year to year) variability of surface mass balance and to extract the part of mass change which is a consequence of meteorological forcing. However, this is only given if the analyses follow a certain quality standard. In terms of unexplainable differences between the methods, a thorough uncertainty assessment has to be conducted in order to indicate that available glaciological balances are questionable and geodetic data can help in identifying shortcomings in the glaciological measurements.

We agree on the limited expressiveness of annual comparisons between the two methods but we show that if analyses are carried out thoroughly significant differences between the mass balance methods are detectable even on the annual scale. We have highlighted and discussed this issue in the revised manuscript.

**1.2 Major comments by referee II (Anonymous Referee)**

**1.2.1 Streamline introduction**

*Streamline introduction - I found the introduction of the paper to be too long and lack appropriate focus for what comes next. While many of the points brought up in the paper are important, they have already been stated in many previous papers. The point (I think) is to see how well geodetic and traditional mass balance methods compare over a suitably long period of time (decade). Perhaps focus on the point that analysis over shorter intervals may miss important processes that reveal themselves for longer periods. At the top of page three we first learn where the paper is going. Please state your objectives earlier and reduce introduction by about 50%. A reader should know at about page 1.5 where we are heading.*

We mainly agree with the argumentation of the referee. The revised introduction is significantly shorter than the original one. We changed the introduction section with the aim of clearly showing the background, motivation and starting point of the paper. This was reached by omitting passages containing information which is common knowledge within the community or which is not relevant for the reader in this part of the paper. Thereby the main objectives of the study are presented earlier in the paper. We also tried to sharpen the motivation of the paper by pointing out the research focus more clearly.

**1.2.2 Reorganization required**

*Reorganization required – I appreciate the detailed attention that the authors pay to processes that could make traditional and geodetic methods differ, but the current organization of these sections comes after key equations used to convert volume change into mass (w.e.) change. You really should present sections (4.1, 4.2 ... ) before you present equations (1) and (2). This is especially evident when one reads section 4.2 and then needs to consider whether equation (5) really differs from equation (2) – it doesn't really. This change would make your paper easier to read (certainly more logical).*

Indeed the structure of the paper was a point of long discussions between the authors. However, we re-structured the paper by shortening the introduction and shifting some of the former content to the beginning of Sect. 3. Furthermore we fully revised Sections 4 and 5 which makes it much easier for the reader to follow the paper.

Apart from that, our (revised) paper is presented following a commonly used structure (Introduction, Study Site, Data and Methods, Results, Discussion, Conclusion...). However, we do not see a logical way to present method inherent differences (sections 4.1, 4.2...) before explaining the methods themselves. Equations 1 and 2 are both fundamental for the geodetic method presented in section 3.2. Only after this method is presented/discussed (Sect.3) there can be a discussion about it (Sect. 4).

Note that we revised Equation 5 and we omitted the conversion factor $K$. However, Equation 5 now specifies how the conversion density $\overline{\rho}$ (which is part of Equation 2) is calculated in our paper (Sect. 4.2). The equation clearly differs from Equation 2.

**1.2.3 Spatial noise**

*Spatial noise – On page 7 the authors discuss using SDz from stable control area to define spatial variability, but I don´t understand how this would yield that information. These control patches serve as so-called "check points" used in traditional photogrammetry. What would they tell us about spatial variability and how it might affect their results? Not much I'm afraid. What would yield that information, however, is the decorrelation length inherent in their data. The authors have gridded data where they can correct their sample sizes for spatial autocorrelation. You should assess the degree of spatial correlation of your data and reduce number of independent samples accordingly. There are several key papers on this topic, one of them (Rolstad et al., 2009) is cited below.*

We did not intend to show the spatial variability of the DTM errors, but to give a measure on the overall DTM accuracy affecting the geodetic mass balances. Since the errors are quite low and do not show large spatial variation within the DTM, this was deemed a comprehensible approach. In our case of ALS (> 1 point per m$^2$) it can be assumed that the number of independent items is about the number of glacier pixels (cf. Joerg et al. 2012). However, since both reviewers criticized this, we changed the way in which we calculate the random error of the used DTM. This also leads to a much lower random error in the DTM. Although we are aware that there is also the implicit assumption that the DTM uncertainty over stable terrain is representative for the DTM uncertainty over the glacier (cf. Rolstad et al., 2009), we did not correct our sample sizes for spatial autocorrelation, but added a clearer discussion of this issue to the revised manuscript.

**1.2.4 Dimensionless conversion factor K**

*Dimensionless conversion factor K - I have a few problems with the introduction of this variable (K) into the literature. First, this is something that is routinely applied in sequential DEM differencing in many previous studies even though it isn´t always stated as such. Second, unless I´ve missed something K should range between 0-1 yet it is state as ranging between 820-930 (line 267). Third, on lines 386-387 the authors state that their new dimensionless conversion factor K now has units of kg m-3. Many have used this conversion factor in past studies; it´s not new, so please let´s not*

*re-invent the wheel and muddle the literature with new dimensionless numbers.*

We omitted this misleading factor from our paper. Equation 5 was revised now explaining the calculation of the mean glacier density $\overline{\rho}$ used for density conversion.

**1.2.5   Clearer explanation and discussion of uncertainties**

*Clearer discussion needed for explaining discrepancies - One of the major conclusions of this paper is that based on the geodetic balance calculations the authors feel that the years 2002/3, 2005/6, and 2006/7 are biased in the traditional mass balance data. I think they are trying to state that the glacier lost most of its accumulation area and the bias was caused by having no stakes high up on the glaciers (in this case probing and pits would yield nothing). This point isn´t as clear as it need to be in lines 410-444; they need to shorten this section, explicitly implicate the methodological factors that could account for the error and then implicate meteorological factors. As it stands they start with the latter without a clear discussion of the former.*

We agree with the referee. In the revised manuscript we attribute the differences between the mass balance methods more clearly to an insufficient measurement set-up and missing observations in the former accumulation area. Furthermore we shortened and restructured Section 5 which results in a more logical stream flow of discussion which is easier to follow for the reader.

**1.2.6   Overly bold statements**

*Avoid overly bold statements - A minor point, but it is best to avoid absolute statements in papers. The authors suggest that their study is the first to compare annually-resolved geodetic and traditional mass balance records, yet a quick literature search indicates that this isn´t correct. For example, Beedle et al. (2014) did this for a shorter period of record and Krimmel (1999) did this for a longer period of time. You should either modify your statements to reflect that your comparison exceeds those of other studies or simply drop statements like this. My preference would be to do the latter.*

We omitted such statements from our paper. Wherever needed we specified the innovative aspects of our study more precisely and we added appropriate references.

**2 Author's response to specific comments by referee I (M. Zemp)**

*Page 3, Line 67, "first use of annual geodetic records": At South Cascade Glacier, annual results from both geodetic and glaciological methods have been analysed by Krimmel (1999): Robert M. Krimmel (1999) Analysis of difference between direct and geodetic mass balance measurements at south cascade glacier, washington, Geografiska Annaler: Series A, Physical Geography, 81:4, 653-658.*
In the revised manuscript we avoid/correctly specify such statements (as also suggested by referee II) and we added references on previous studies.

*P4, L119, "Results are submitted to the WGMS ...": you could add a reference to WGMS (2017, and earlier reports): WGMS 2017. Global Glacier Change Bulletin No. 2 (2014–2015). Zemp, M., Nussbaumer, S. U., Gärtner- Roer, I., Huber, J., Machguth, H., Paul, F., and Hoelzle, M. (eds.), ICSU(WDS)/IUGG(IACS)/UNEP/UNESCO/WMO, World Glacier Monitoring Service, Zurich, Switzerland.*
In the revised manuscript we refer to WGMS (2015, 2012, and earlier volumes).

*P6, L178-184, Equation 2: the geodetic balance is usually calculated using the average glacier area of the two surveys (Zemp et al., 2013, The Cryosphere, Eq. (5) and (6)). At annual time steps, this might not make a big difference, but for the decadal period with a surface area reduction of 15% it does become relevant.*
We agree. The calculation of geodetic mass balances was adapted now using the time averaged area mean $(S_{t0} + S_{t1})/2$.

*P6, L188 & Fig 1, stable areas: I fully support the decision to complement the down-valley soccer field with stable areas near the glacier. Please add a short comment about the selection criteria for the stable areas A-E.*
The selection is based on visual inspection and expert knowledge about the terrain (Sailer et al., 2012; Bollmann et al., 2011). A respective comment was added to the revised manuscript.

*P8/9, L240-267, density conversion: the density conversion factor depends on changes in the three-dimensional firn body and is a function of (i) the additional snow layer incl. related densification and metamorphosis, (ii) firn compaction and metamorphosis, and (iii) sub/emergence velocity. From the text, I cannot fully comprehend how these factors are covered (or not) by the author's approach combining differential DTMs, surface classifications, and density assumptions. Please clarify and discuss the opportunities and limitations of the used approached.*
The conducted density conversion consists of three steps within our approach. First, the dDTM was calculated. In a next step, the glacier surface was classified into two classes (firn and ice) by using the intensity images of the ALS campaign, resulting in surface grids for each year. By subtracting the classified intensity rasters and reclassifying the resulting new surface raster, we incorporated the changing extent of the perennial firn zones in a third step. This should answer point (ii) raised by the reviewer. However, we are aware that firn compaction and metamorphosis are not covered by this approach. Point (iii) could not really be considered using the available data, which is why we already mentioned in the introduction that we will not incorporate glacier flow dynamics in the presented analysis. Regarding point (i), the snow layer was incorporated by combining a maximum snow height at the time of measurement with in-situ measured snow-densities, to redistribute the mass according to the snow layer to the glacier surface. Nevertheless, we are aware that this type of spatial distributed density conversion is rather a best guess than a three-dimensional modelling of the firn body. The revised manuscript contains a fully revised Section 4 where we tried to clarify and better discuss our way of density conversion. In the revised text we refer to Figure S1 which was added to the Supplement of the paper (see Fig. 1). This helps the reader to better follow our analysis.

[Figure]

Figure 1: Flow chart illustrating the scheme of density calculation.

*P9, L266-267 & Table 5, density conversion factor and related uncertainties: for a non-expert it is hard to follow how the density conversion factor and corresponding random uncertainties (together with the annual balance) relate to K.sigma and K.epsilon in Table 5. Adding a corresponding equation in Section 4.2 might help.*

The misleading density conversion factor $K$ was omitted from the revised manuscript. Equation 5 was changed accordingly. The revised symbology of uncertainties, as well as their calculation, now is in line with Zemp et al. (2013). We added a respective statement at the end of the revised introduction.

*P9, L271, "stratigraphic year": I think this should be "end of the hydrological year" or "fixed date system" (cf. P9, L275, "30th September").*
Changed accordingly to "end of hydrological" year

*P10, L285-287, "elevation dependent mean ablation gradient": do you use the same gradient for the ablation and the accumulation zone? Please clarify.*
Yes we do. This is explicitly mentioned (and justified) in the revised manuscript.

*P11, L323-324: for comparability, convert the values by (Thibert et al., 2008) to annual change rates.*
Done.

*P15, L448-251, "were the first and so far only": consider rewording in view of earlier studies at South Cascade by Krimmel (1999, Geogr. Ann.).*
The sentence was changed and the reference was added to the paper.

*Text, Figs & Tabs, "altitude" versus "elevation": in most cases, you could replace "altitude" by "elevation" (cf. McVicar, T. R., & Körner, C. (2013). On the use of elevation, altitude, and height in the ecological and climatological literature. Oecologia, 171(2), 335-337.)*
 Done as far as possible.

*P24, Fig. 1: For clarification, you could write in the figure caption: "Note that in 2003, no accumulation measurements COULD have been carried out DUE TO THE STRONGLY REDUCED ACCUMULATION ZONE. HENCE, only ablation stakes were available."*
Done.

*P25& 30, Fig. 2 & 7: the two figures are redundant to a certain degree. On the other side, it is not fully clear, which differences and uncertainties are included. Please at least clarify in captions. In addition, you could consider merging Fig 2 & 7, showing bias corrections for both glaciological and geodetic results. Instead, you could remove the cumulative curves (=¿ shown in Fig. 8).*
Figures and their captions were revised and their number was reduced. We omitted the bias correction from the revised manuscript since the analysis of reduced discrepancies between the two mass balance methods does not justify such.

*P26. Fig. 3: I would add a bar showing the intensity range (values) to the legend of the left image. In the legend of the right one, I would replace "perennial firn" by "snow and firn".*
**Done**

*P27, Fig. 4: In the caption, please clarify what you mean with "Corrected". It might be sufficient*

*adding a reference to the corresponding section in the paper. I would add the terms glaciolocial and geodetic to the label of the x-axis in the left and right figure, respectively. In addition, please add a note on the effect of the sub/emergence velocity.*
**Done**

*P28, Fig. 5: you could add the data point(s) for the full period (glaciol.cum versus geod.cum, glaciol.cum versus geod.01/11).*
Figure 5 was omitted from the revised manuscript since it was redundant to Figure 2.

*P29, Fig. 6: please add a note on the effect of the sub/emergence velocity.*
The following note on the influence of glacier dynamics was added: Note that vertical profiles of the two methods cannot be directly compared due to the effect of glacier dynamics which leads to more negative geodetic results (than the glaciological ones) in the higher elevated areas and vice versa in the lower glacier regions.

*P31, Fig. 8: typically, one would calibrate the glaciological with the geodetic over the decadal period (i.e. 2001-11). Hence, it might be good to show that result here too.*
Since the results of the two methods agree well on the decadal time scale, a calibration is of the glaciological series is not justified and was hence omitted from the revised manuscript.

*P34, Tab. 2: you could add a column for the two dDTM of the full period, i.e. 01/11.*
Done. However, this table was shifted to the Supplement in the revised paper version.

*P35, Tab 3: please explain why the density given in the cation ($900 \, kg \, m^{-3}$) differs with the one mentioned in the text ($850 \, kg \, m^{-3}$, cf. P8, L249)*
The set of tables was reduced and fully revised. Table captions are also revised and the mentioned typo and was clarified.

*P36, Tab 4: in the caption, there are some problems with the symbol for average SC. What is the "mean acc. area"? Do you refer to the end-of-summer accumulation area?*
Symbol problems have been revised. Mean accumulation area is the classified firn area (AF). To avoid ambiguity it was changed.

*P37, Tab 5: I would expect the annual uncertainties for the density conversion (sigma K) to be larger than for the (zonal) ones for the ALS-DTM (sigma DTM)... see also my comments above (substantial point (a) and comment related to density conversion, P9, L266-267)*
Since we changed the calculation of the errors for the ALS-DTM (sigma DTM), those are now lower than in the originally submitted manuscript.

*P38, Tab 6, caption: consider rewording "improved balance" into "bias-corrected balances"; consider rewording "statistical significance" by "reduced discrepancy". Use the same symbol for the common variance in caption (now wrongly epsilon.comvar) and table (=> sigma.comvar).*

Done

**3 Author's response to specific comments by referee II (Anonymous Referee)**

*A clunky title. I'd suggest. 'Geodetically corrected (or Homogenized) mass balance series of Hintereisferner Glacier, Austria for the period 2001-2011'*
The revised manuscript is entitled: *"Geodetic reanalysis of annual glaciological mass balances (2001-2011) of Hintereisferner, Austria"*. The revised title is more specific and less clunky.

*First sentence needs to be reworded. It sounds like you obtained 2001-11 mass bal- ance(s) records ...*
The sentence was changed.

*Line 18: Sentence needs revision (grammatically incorrect)*
The sentence was revised by a native speaker.

*Line 23: Replace 'as a substitute for' with 'superior to'*
The sentence was revised.

*Line 39: Delete 'and within the snow' since the top of this layer defined glacier surface by definition.*
Done.

*Line 40: Replace 'subtracts' with 'differences'*
Done.

*Line 45: Full stop missing after 'glacier'*
Full stop added.

*Line 50: Beedle et al. (2014) is missing from this list*
The reference was added.

*Line 63-66: Confusing and poorly worded sentences. Please revise.*
The section was revised.

*Line 67: See major comment (F)*
We avoided such statements in the revised manuscript. Please also see our detailed reply to the respective major comment.

*Line 86: Add 'an average' after 'with' and strike 'in average'*
Done.

*Line 86: What is a 'totalizing rain gauge' - bulk collector?*

We use the term totalizer rain gauge as defined in the meteorology glossary by the American Meteorological Society. See $http://glossary.ametsoc.org/wiki/Totalizer\_rain\_gauge$.

*Lines 90-94: Tangential to paper's focus (delete).*

Done.

*Lines 100-101: Add 'Annual' at start of sentence, strike 'annual mass balance' and 'have been started' and replace with 'commenced' and strike 'are carried out regularly since then'*

Done.

*Line 129: Strike 'among others' - meaningless in its usage here.*

Done.

*Line 132: 'Further explanation...' - Unclear why this statement is here. Reads like an orphaned one.*

The sentence was deleted.

*Line 143: replace 'wrong' with 'incorrect'*

Done.

*Line 148: 'For extrapolating ... ' - This sentence is linked to nothing (a single thread). Not sure why it is here.*

The sentence was deleted. The whole section has been revised.

*Line 154: Replace 'according to the law of error propagation' with 'by error propagation' - There are few physical laws.*

Done.

*Line 180: I had commented in the paper margin 'are density differences treated per elevation band' and hence my suggestion for you to move sections 4.1 and 4.2 before equation (1). See major point (B).*

We refer to our detailed response in Sect.1.2.2 of this document.

*Lines 198-200: See major point (C ).*

We refer to our detailed response in Sect.1.2.3 of this document.

*Lines 212 (and throughout paper): Try not to state things like 'Figure 2 shows... '. State trend, ob-*

*servation and refer to figure at end of the sentence. For example, 'Density increases with elevation (Figure 2)'. This allows reader to digest your point and then refer to figure (it also reduces verbiage).*
We avoid such throughout the revised manuscript.

*Line 220: Move 'significantly' before 'influence'*
Done.

*Line 228: Add 'absolute' before 'vertical' and strike vertical lines as they are impossible to see in running text.*
Done.

*Line 231: Strike 'very' and avoid this vague qualifier at all costs.*
Done.

*Line 265: See point (D).*
We refer to our response in Section 1.2.4 of this document.

*Line 271: Replace 'a multi methodical approach was applied incorporating' with 'we incorporated'*
**Done.**

*Line 276: remove (s) from extrapolations*
Done.

*Line 280-281: How does this standard lapse rate compare to one assessed with station data. Does this help to explain differences in the extreme melt years?*
In the case of significant snow fall we assume the atmosphere to be saturated which justifies the use of the moist adiabatic lapse rate. This is supported by analysis of station data which shows that the use of this gradient in such cases is quite a reasonable assumption. However, the large differences between the two mass balance methods are not sensitive to the choice of temperature lapse rates which, hence, do not significantly contribute to explaining those differences.

*Line 291: Replace '5' with five. Write out all numbers less than 10 unless number has a unit. For example, seven stakes but 3 cm.*
Done.

*Lines 305-212: So how does this approach potentially affect your results? So if you simply ignored effects of crevasses what would results show?*

The negligibly small effect of crevasses is discussed in detail in the revised manuscript (Sections 4 and 5).

*Line 326: Stylistic point, but 'frictional dissipation' I believe is the more precise term.*
Changed.

*Line 348: Write out 'E.g. - Never start a sentence with this.*
Changed.

*Line 353: If it's a small term at the annual scale, it's small and within error at decadal scale. It can't be significant for one but not the other. Suggest dropping last clause in sentence.*
Changed.

*Line 357: 'The 2001 to 2011 one step...' - Not sure what sentence is trying to state.*
Mass change over the whole 10-year period can be calculated either by summing up the results for individual years or by calculating $\Delta V_{2001-2011}$ by differencing the two DEMs of those years and converting this volume change to a change in mass (the latter is the *one-step* analysis). The results for both ways are slightly different due to density issues and different reference areas. However, in the revised manuscript we only show the 'one-step analysis' since the cumulative geodetic balance does not add any value but causes confusion.

*Line 358-359: See earlier comment about 'Table and Figure shows...'*
Changed.

*Line 363: What does 'respectively' refer to? The penultimate sentence? Revise.*
Revised.

*Line 370: Sometimes last word before equation has a colon sometimes not, be consistent with journal standards.*
Revised according to author-guidelines of TC.

*Line 371: How does δ change if you incorporate effective degrees of freedom in the geodetic estimate of uncertainty (i.e. correct for spatial autocorrelation)?*
We kindly refer to our detailed response in the general comments (Section 1.2.3 of the present document).

*Line 376: Replace 'coherent' with 'similar'*
Done.

*Line 378: How is bias defined in this paper? Should be formally defined.*
The introduction of the revised manuscript contains a statement pointing out that our analyses and interpretations follow the guide lines of Zemp et al. (2013). This is also valid for the term *bias* which is used as a synonym for systematic errors.

*Line 383: Did you really explore the parameter space? This phrase is typically used with Monte Carlo sampling or Latin Hypercube sampling. Did you do that?*
No. We rephrased this section in the revised manuscript.

*Line 388: Why does K now have units? You told us earlier that it was dimensionless...*
K was omitted from the paper. See above statements.

*Line 400: If you used results that weren't smoothed (removal of crevasses) how does this affect your results?*
This (small) impact is discussed in the original and the revised paper version.

*Lines 410-440: I found this portion extremely difficult to follow for reasons outlined in major point (E).*
We refer to our response in Sect. 1.2.5 of this document. The paper section of concern was revised.

*Line 448: See point (F). Several papers out there that do this. Your paper, however, does this for the longest series, and will be well received. But please don't oversell its novelty.*
We avoid such statements and added references in the revised manuscript. We refer to our response in Sect. 1.2.6 of this document.

*Line 452: Change sentence to, 'It neither include(s) a through...nor '*
Sentence was revised.

*Line 453: Remove 'ed' from showed.*
Done.

*LIne 456: Change 'a snow cover' to 'snow'*
This sentence was rephrased.

*References: I did not check these for typos, but suggest you add the ones in this review to the list.*

Done.

*Figures:*

*Fig. 2 - A legend added to this figure would help reader. It would be nice in the figure caption to state level of uncertainty (68, 95%).*
Done.

*Fig. 3 - What are units of Intensity (DN?). $kg\ m^{-3}$.*
Intensity has no units. The used intensity rasters only show the backscattered energy stored in 8 bit or 256 grey values.

*Fig. 4. - You don't deal with dynamics (flux divergence) so it is not appropriate to plot these data as 'Mass balance [m w.e.]' as a function of elevation. $\frac{dh}{dt}$ and b are not equal due to dynamics. This plot must be redrafted showing 'Elevation change [m w.e.] and not 'Mass balance'.*
It is true that we do not explicitly resolve ice dynamics. Nevertheless, the change in surface elevation can be used to calculate a mass balance which is the result of accumulation/ablation processes (surface, internal, basal) and ice flux divergence. In principle it does not matter if this is done for a point/column, an elevation band or the whole glacier (e.g. Cogley et al., 2011, page 5). However, we think that comparisons of geodetic and direct balances are only problematic if not done on the glacier wide scale. Hence, we did not redraft the plot but added a statement (as suggested by referee I) discussing the effect of ice dynamics on the local mass balance and the implications on method-comparisons on scales others than "glacier-wide".

*Fig. 6. Remove titles from figures and simply use 'a)' and 'b)' . Avoid excessive qualifiers. Change 'The extraordinary mass' to 'Mass'*
Done.

*Fig. 9. Is this really the best way to show these data? Why not simply remove the figure and tell reader in text if homogenized hintereisferner series correlates more strongly (or use of other statistic than Pearson) with nearby series.*
This figure was moved to the supplementary material. The text section relating the mass balance of Hintereisferner to those of nearby mass balance glaciers was shifted to the supplementary as a caption of figure S3.

*Tables:*

*There are a lot of them and not sure if they are all needed. Any individual wanting your data would request them, no? Alternatively you could deposit them with the WGMS or other agency (or include*

*as electronic supplementary data). They take up a lot of journal space and some repeat what figures show.*
We reduced the amount of figures and tables and shifted some of them to the supplementary material.

*Table 5. Replace 'cum' with 'Sum'.*
Done.

15 Before approaching the reanalysis of the annual surface mass balances of Hintereisferner for the time period 2001 to 2011 further uncertainties in the glaciological mass balances series must be addressed. The glaciological method suffers mainly from uncertainties related to (i) point measurements and (ii) their spatial extrapolation over the entire glacier (e.g., Zemp et al., 2013; Galos et al., 2017). For both uncertainty sources and due Due to the lack of respective data on Hintereisferner we synthesize appropriate information from the literature as follows. (Zemp et al., 2013) analysed , among others, to estimate both sources of uncertainty. Zemp et al.

20 (2013) analysed the mass balance series of Hintereisferner for six periods between 1953 and 2006 and attributed an uncertainty of $\pm0.10$ m w.e. $\mathrm{a^{-1}}$ to field measurements for the years after 1964 and doubled the value for the years before. For the spatial interpolation of point data they assigned values between $\pm0.14$ and $\pm0.54$ m w.e. $\mathrm{a^{-1}}$ with an average of $\pm0.33$ m w.e. $\mathrm{a^{-1}}$ for the entire period. Further explanations are not provided by Zemp et al. (2013).Fountain and Vecchia (1999) found combined uncertainties for (i) and (ii) of up to $\pm0.33$ m w.e. $\mathrm{a^{-1}}$ by analysing the modelled variability of the mass balance of South Cascade glacier.

25 Thibert et al. (2008) and Thibert and Vincent (2009) analysed 51 years of mass balance for Glacier de Sarennes and reported a combined annual uncertainty of $\pm0.20$ m w.e. $\mathrm{a^{-1}}$ for (i) and (ii). For Gries- and Silvrettagletscher, Huss et al. (2009) assumed overall uncertainties related to (i) and (ii) of $\pm0.16$ to $\pm0.28$ m w.e. $\mathrm{a^{-1}}$ . By investigating the glaciological and geodetic mass balances of Storglaciären, Zemp et al. (2010) determined the random uncertainty for (i) and (ii) with $\pm0.10$ m w.e. $\mathrm{a^{-1}}$ each, which resembles the results of Jansson (1999). For Findelengletscher, Sold et al. (2016) roughly estimated a random uncer-

30 tainty of $\pm0.04$ m m w.e. $\mathrm{a^{-1}}$ for (i), referring to Huss et al. (2009), and of $\pm0.17$ m w.e. $\mathrm{a^{-1}}$ for (ii) by evaluating contour lines drawn independently by 18 analysers. On Nigardsbreen, Andreassen et al. (2016) obtained a total point measurement uncertainty of $\pm0.25$ m w.e. $\mathrm{a^{-1}}$ as the root sum square (RSS) of a false determination of the previous year's summer surface ($\pm0.15$ m w.e. $\mathrm{a^{-1}}$), upwelling up-welling of stakes ($\pm0.20$m w.e. $\mathrm{a^{-1}}$ ), and wrong incorrect density assumptions of snow and firn ($\pm0.05$ m w.e. $\mathrm{a^{-1}}$). Uncertainty of spatial integration was taken as $\pm0.21$ m w.e. $\mathrm{a^{-1}}$, made up by point measurements

35 insufficiently covering both the vertical range and the total area of the glacier.

Based on the findings of Zemp et al. (2013) combined with expert knowledge about the study site, we assess the uncertainty related to point measurements at Hintereisferner, being to be in the order of $\sigma_{point}$ $\sigma_{glac.point}$ = $\pm 0.10$ m w.e. a$^{-1}$, resulting in a decadal value of about $\pm 0.32$ m w.e. For extrapolating point data into reasonable patterns of mass balance, the contour line method uses expert knowledge. Based on Sold et al. (2016) , we estimate a respective uncertainty Hintereisferner we estimated the uncertainty related to extrapolation of point data based on Sold et al. (2016) leading to an annual value of $\pm 0.15$ m w.e. a$^{-1}$ for Hintereisferner. In addition and according . Additionally we accounted for the presence of large areas not covered by point measurements. According to Andreassen et al. (2016), we assume that the extrapolation over areas not covered by point measurements inherits those areas inherits further uncertainties of $\pm 0.10$ m w.e. a$^{-1}$. Hence, the uncertainty due to spatial integration of the respective measurements over the entire glacier is defined to be $\sigma_{spatial}$ $\sigma_{glac.spatial}$ = $\pm 0.18$ m w.e. a$^{-1}$ and result in decadal uncertainty of the related decadal uncertainty is $\pm 0.57$ m w.e.. Overall uncertainties for the glaciological mass balances are calculated, according to the law of error propagation following Zemp et al. (2013, Eq. 14), leading to $\sigma_{glac}$ an annual value of $\sigma_{glac.total}$ = $\pm 0.21$ m w.e. for annual and which corresponds to a cumulative uncertainty of the glaciological method (2001 to 2011) of $\pm 0.65$ for the cumulated values m w.e.

**3.2 The geodetic method**

Between 2001 and 2011, eleven 11 ALS flight campaigns had been carried out near the end of each mass balance year (see Table 1 Tab. 1). During each ALS data acquisition campaign, the glacier was covered with a number of overlapping flight strips in order to increase the point density and to ensure high quality and complete coverage of the glacier (Wever and Lindenberger, 1999; Geist et al., 2007). As there is essentially no high vegetation in the study area, ALS points are classified into ground points and flying objects (outliers) only. The ground points of all datasets are imported into a laser database system (Rieg et al., 2014) which facilitates storage and further processing. DEMs of 1 m resolution DTMs are were calculated for all datasets, whereby the mean value of all ALS points located in each cell represents the elevation of the cell. The elevation values for the few raster cells that do not contain a single point are interpolated from the neighbouring cells using a least squares method. In order to provide high-quality DTMs DEMs used for mass balance calculations, horizontal misalignment of the DTMs DEMs being differenced has to be excluded. Therefore a statistical co-registration correction procedure as suggested by Nuth and Kääb (2011) was performed for this study. Following Joerg et al. (2012) we applied the first two steps of the procedure to the ice-free areas for identifying potential horizontal shifts and vertical offsets between two ALS-DTMs ALS-DEMs. The statistical co-registration reveals horizontal shifts smaller than the DTM DEM pixel resolution with no elevation-dependent bias, and the DTMs hence, the DEMs can be subtracted from each other without performing DTM DEM corrections. The total volume change $\Delta V$ between two dates is then derived from the respective elevation difference $\Delta h_k$ of the two grids at pixel $k$ with cell size r of the DTMs DEMs, summed over the number of pixels $K$ covering the glacier at the maximum extent and is expressed as (cf. Zemp et al., 2013) follows (cf. Zemp et al., 2013):

$$\Delta V = r^2 \sum_{k=1}^{K} \Delta h_k,.$$
(1)

For a comparison with the glaciological balance, $\Delta V$ is then converted into a specific mass balance in the unit meter water equivalent (m w.e.):

$$B_{geod} = \frac{\Delta V}{1/2 \cdot (S_{t1} + S_{t2})} \times \frac{\Delta V}{1/2 \cdot (S_{t0} + S_{t1})} \cdot \frac{\overline{\rho}}{\rho_{water}}, \qquad (2)$$

where $S_{t0}$ and $S_{t1}$ are the glacier areas at the first ($t_0$) and second ($t_1$) acquisition date respectively and $\overline{\rho}/\rho_{water}$ is the ratio between the average bulk density (see Eq. 5 in Sect. 4.2) of $\Delta V$ and the density of water, at the first acquisition date $t_1$..

Despite a thorough co-registration, surface elevation differencing of two DTMs DEMs is still subject to various uncertainties. The vertical accuracy of the raw ALS point data was first assessed by comparing the point clouds with differential global navigation satellite system (dGNSS) measured points on a homogeneous horizontal surface outside the study area (e.g. in our case a football field in Zwieselstein 20 km down-valley of Hintereisferner). Table 1 shows the The standard deviations (SD) of vertical accuracies of the individual datasets are shown in Table 1 t1. As the reference surface does not reflect the surface conditions in terms of slope, aspect and roughness, and therefore is not representative for vertical accuracies, Bollmann et al. (2011) Bollmann et al. (2011) compared dGNSS ground control points with laser returns (deviation to laser points 0.07 m, standard deviation 0.08 m) and calculated an absolute slope-dependent vertical accuracy for Hintereisferner ALS point data ($<0.10$ m on slopes $<40°$). Sailer et al. (2014) analysed the uncertainties resulting from rasterizing laser point clouds, revealing that a cell size of 1x1 m as used for our study causes only negligible errors of less than 0.10 m. For the raw geodetic balance ($b_{geod.raw} B_{geod}$), the results of DTM DEM differencing over stable terrain are taken to define uncertainties associated with the DTM DEM comparison. Therefore, we selected five 5 stable control areas ($3 \times 10^4 \ m^2$ $3 \times 10^4$ m$^2$) surrounding the glacier (Figure 1 Fig. 1), in order to quantify grid-based uncertainties of spatially averaged elevation differences. As The selection of these sites is based on visual inspection and expert knowledge about the terrain around Hintereisferner (Bollmann et al., 2011; Sailer et al., 2012). According to (Rolstad et al., 2009), we assumed that the DEM uncertainty over stable terrain is representative for the entire glacier. However, we did not correct our sample size for spatial autocorrelation, but, following (Joerg et al., 2012), we assumed that the number of independent pixels is about the number of glacier pixels and used the standard error as uncertainty measure. Thereby the standard deviation of the elevation differences ($SD\Delta_z$, Table 2) provides information on the spatial variability of the selected stable areas, we used the related RSS for an approximation to our DTM uncertainty : Tab. S1 in the Supplement) provides the basis for assessing the influence of random pixel-elevation uncertainty on the glacier wide geodetic balance $\sigma_{DEM}$:

$$\sigma_{DTM \ DEM} = \sqrt{\sum_{1}^{i}(SD_i^2) \frac{\sum_{i=1}^{n}(SD_i)}{\sqrt{n}}}, \qquad (3)$$

where SD is the standard deviation within the reference surfaces i. The result was converted into mass using the density of ice. Comparison of the differential DTMs (dDTMs) show uncertainties of $\pm 0.06 < \sigma_{DTM} < \pm 0.17$ $i$ and $n$ is the number of pixels in stable areas. This procedure yields uncertainties of $\pm 0.012 < \sigma_{DEM} < \pm 0.024$ m w.e., resulting in and $\sigma_{DEM} = \pm 0.36$ 0.087 m w.e. cumulated over the observation period (01-11$_{cum}$; Table 3). In contrast, for the 2001 to 2011 one step application of the geodetic method (01/11; Table 3) yields a value of $\sigma_{DTM} = \pm 0.14$. Table 3 summarizes the results of sections 3.1 and 3.2 and shows the differences between the adjusted glaciological and the raw geodetic mass balances ($b_{glac.hom}$ - $b_{geod.raw}$ analysis (Tab. 3).

**4 Accounting for method Method inherent differences**

Figure 2 shows The differences between the glaciological and the geodetic mass balance series as revised in sections 3.1 and 3.2. The expected differences vary from year to year, being particularly high in some years (Table 3 certain years (Fig. 2, Tab. 2). The potential causes of these discrepancies in the mass balance series are related to a number of factors: snow cover at the time of ALS acquisition(4.1), different glacier-wide density assumptions in mass balance calculation(4.2), survey date differences between the glaciological and geodetic observations(4.3), the way the methods consider the existence of crevasses (4.4), and the differences between the surface (glaciological) and the total (geodetic) mass balance (4.5) and the different processes captured by the two mass balance methods. All those issues are thoroughly assessed below.

**4.1 Differences induced by snow cover present in DTMsDEMs**

Whereas the vertical accuracy tends to be very of ALS-DEMs is high, biases as a result of snowfall events preceding the ALS surveys significantly influence the calculated volume change significantly. From the analysis of elevation differences in the non-glaciated terrain, the mean difference between two DTMs ($\overline{\Delta Z}$) stable areas ; Table 2) with DEMs in stable areas ($\overline{\Delta z}_{stable}$) can be used to correct for DEM-biases ($\epsilon_{DEM}$) caused by the presence of snow as follows:

$$\epsilon_{dtm\,\text{DEM}} = \frac{\sum_1^n \overline{\Delta Z_l}}{n} \frac{\sum_{i=1}^{n} \overline{\Delta z_i}}{n}, \tag{4}$$

where n is the number of DTM DEM grid cells covering stable and non-glacierized terrain, can be used for inevitable volume corrections, caused by preceding snow fall events. For the periods 2001/02, 2005/06, 2006/07 and 2007/08 the investigation of stable areas within the dDTMs dDEMs revealed snow induced absolute vertical offsets between (0.18 ) and (and 0.58 ) m ($\overline{\Delta Z}$); bold numbers in Table 3 m (see bold numbers for $\overline{\Delta z}_{stable}$ in table S1 of the Supplement). In all other dDTMsdDEMs, the vertical bias was below 0.10 m. In 2004 and 2010 a snow fall event occurred some days before the ALS measurements. However, this is not reflected in the stable areas of the respective dDTMdDEM, because the snow in non glacierized areas had melted from off-glacier surface by the time of the ALS survey. This leads to a very low small offset in the non-glacierized terrain in the related mass balance periods. Yet, as snow cover increases, the ALS elevations measured on reference surfaces have to be cross-checked with snow depth data from the closest field survey data for snow depth estimation and subsequently and subsequently they have to be corrected. Based on the altitude distribution of stable areas and in-situ measurements a linear regression in 50 m elevation bands yielded yields mean snow depths of 0.52 m in 2001, 0.23 m in 2004, 0.46 m in 2005, 0.13 m in 2006, 0.12 m in 2007 and 0.26 m in 2010. This leads to adjusted DTMs DEMs and, finally, to a respective mass balance correction value $\epsilon_{DTM}$ $\epsilon_{DEM}$ (Table 5). Furthermore this approach was integrated to the estimation of differences related to unequal survey dates (see section Sect. 4.3).

**4.2 Density conversion**

One of the method-inherent differences between glaciological and geodetic method can be found in the density conversion. Glaciological While glaciological mass balances are derived from mass change measurements calculating mass change based on well constrained in situ density measurements , whereas the geodetic ones in-situ measurements of density, geodetic balances are based on volume change measurements, which require

conversion to mass by an estimated densityfor the material lost or gained (e.g. Thompson et al., 2016). volume-to-mass conversion using estimates of bulk density. Several studies assume that density in the accumulation area is constant over time and, hence, use glacier ice density for the conversion (e.g. Andreassen, 1999; Haug et al., 2009). As (e.g., Andreassen, 1999; Haug et al., 2009). But as long as snow or firn is present, the density of ice ($\rho_{ice}$=900 $kg\ m^{-3}$) doing so causes an overestimation of the mass change. Hence, only below the equilibrium line altitude (ELA), where altitudinal changes are either due to ice ablation or emergence, the use of the density of ice is appropriate. However, if only appropriate in glacier areas without firn. If year-to-year firn line changes are known, the volume to mass volume-to-mass conversion can be approximated by improved by using an average density of firn (e.g. Sapiano et al., 1998; Prinz et al., 2011).To make a first calculation of mass change (Figure 2) for changes in the accumulation area (e.g. Sapiano et al., 1998; Prinz et al., 2011), we follow the recommended approximation for density conversion of 850 .

In the present study, ice density ($\rho_{ice}$ = 900 $\mathrm{kg\,m^{-3}}$) was only applied to the ablation areas, where altitudinal changes are either due to ice ablation or glacier dynamics while the geodetic mass change in (perennial) firn areas was calculated using a density of $\rho_{firn}$ = 700 $\pm_{60}$ $kg\ m^{-3}$ suggested by Huss (2013). However, this approach revealed differences in some periods of the data series, as the assumption of Huss (2013) is suitable for geodetic analyses over periods which span over five years or more and which show relatively stable mass balance gradients, non-negligible changes in volume and a relatively stable extent of the firn region.Therefore 50 $\mathrm{kg\,m^{-3}}$ (Ambach and Eisner, 1966; Huss, 2013). Consequently, we calculate the annual conversion density $\overline{\rho}$ as used in Eq. 2 as follows:

$$\overline{\rho} = \frac{\rho_{\mathsf{ice}} \cdot \Delta V_{\mathsf{ice}} + \rho_{\mathsf{firn}} \cdot \Delta V_{\mathsf{firn}}}{\Delta V}, \qquad (5)$$

where $\Delta V_{ice}$ and $\Delta V_{firn}$ are the mass changes in ice and firn areas respectively which both add up to the glacier wide volume change $\Delta V$.

In order to classify the glacier surface into ice and firn zones we designed a pixel-based surface classification workflow, in order to account for changing firn areas. The present classification is work-flow based on ALS-intensity data as described by Höfle and Pfeifer (2007) . Following Fritzmann et al. (2011), a classification of ice and firn zones on the glacier surface for each survey year could be achieved (Figure 3). If no suitable intensity following Höfle and Pfeifer (2007) and Fritzmann et al. (2011) (Fig. 3). This approach was applied to all years with suitable intensity-data while for years when no such data are availablefrom the ALS, the most contemporary ortho-images (e.g. for the year 2010) and/or LandsatTM images (e.g. for the years 2001 and 2004) are were used for surface classification . To incorporate the changing extent of the perennial firn zones we subtracted the surface grids of the respective mass balance periods from each other and reclassified the resulting new surface raster. The glacier surface is classified in two categories: glacier ice with a density of 900±17 kg m⁻3 and perennial firn with 700 ±50 kg m-3 (Ambach and Eisner, 1966; Huss, 2013), whereas the difference to maximum/minimum estimates (±17 and ±50 kg m-3) serve as an uncertainty measure within our approach ($\sigma_K$; Table 5(see Fig. S1 in the Supplement). The resulting grids are used to convert volumetric changes into a massfor every pixel (see equation 2). For a better interpretation we introduce a dimensionless conversion factor as

$$K = \frac{\rho_{ice} \cdot \Delta V_{ice} + \rho_{firn} \cdot \Delta V_{firn}}{\rho_{water} \cdot \Delta V_{total}}.$$

Corresponding volume-to-mass conversion factors ($K$) resultant grids for each survey year were then used for a pixel-based conversion of volumetric changes to changes in mass. Respective values for the conversion density $\overline{\rho}$ lie in the range of 820 to 930 $\mathrm{kg\,m^{-3}}$ and are shown in table 3. Although neither firn processes like compaction or melt water refreezing, nor the impact of glacier dynamics are explicitly resolved our approach is considered to notably improve the quality of our annual results

compared to calculations based on a fixed glacier wide conversion density.

Uncertainties related to density conversion were estimated as follows: $\sigma_{dc}$ was assessed based on the estimated uncertainty ranges of $\rho_{ice}$ and 930. The change in mass balance values compared to the raw geodetic results (Table 2) is ascribed to the density conversion deviation ($\epsilon_K$; Table 5).$\rho_{firn}$ ($\pm 17$ and $\pm 50\ \mathrm{kg\,m^{-3}}$) while $\epsilon_{dc}$ was calculated as the difference between our geodetic mass balance values and those based on a $\overline{\rho}$ of $850\ \mathrm{kg\,m^{-3}}$ as suggested by Huss (2013).

**4.3 Survey date differences**

Temporal differences between the geodetic and glaciological observations need to be addressed. To align the geodetic dates with the end of the hydrological year used for the glaciological mass balance measurements, a multi-methodical approach was applied, incorporating field measurement minutes, DTM analysis results from section 4.1 and data from in situ measurements. Apart from 2011 with in situ measurements conducted on the same day as the ALS flight (Table 4 Tab. S2 in the Supplement), the changes in snow depth and ice ablation between the two measuring dates mass changes during the period between the survey dates of the two mass balance methods have to be considered. If the date of the ALS acquisition deviates from the $30^{th}$ of September (To align the geodetic dates with the end of the hydrological year), the geodetic mass balance is adjusted to the fixed dates by linear extrapolations as follows. In case of ablation between the survey and the fixed date the extrapolation is based on the ablation trend over the immediately preceding time for each stake. This is calculated from used for the glaciological balances and for a corresponding adjustment of the geodetic results we incorporated data from in situ measurements and field work minutes as well as dDEM-based snow cover analysis (Sect. 4.1). Thereby ablation was assessed based on available stake readings during the summer justified by extrapolated air temperature late summer. Observed ablation trends between the observation dates were used to calculate mass change. If necessary, ablation was reconstructed by linearly extrapolating observed trends beyond the stake reading dates. Such cases were cross checked and adjusted based on meteorological data from Vent allowing ablation conditions. In the case of accumulation between the survey. The linear regression of point ablation versus altitude was finally used to calculate spatially extrapolated ablation. Note that the same altitudinal ablation gradient was used for the whole glacier since considerable ablation is restricted to the lower glacier part in this time of the year.

Accumulation between the ALS-survey and the fixed date, the precipitation gradient was assessed based on recorded precipitation at Vent which was extrapolated to the glacier applying observed long-term precipitation gradients between Vent and five 5 rain gauges in the Hintereisferner basin (Figure 1) is used for adjustment to the fixed date Fig. 1). The snow-rain threshold of 0°C is obtained from the Vent temperatures along a lapse rate of $0.0065\ \mathrm{{}^\circ K\,m^{-1}}$.

The survey date adjustment is performed individually for each annual geodetic mass balance, dependent on the presence/absence of snow during the field survey and the ALS data acquisition as well as on the difference between the survey dates and the end of the hydrological mass balance year. Accordingly, we proceeded as follows:

i) If there was no snow cover during both surveys, and the ALS campaign took place before the field survey, an elevation dependent mean ablation gradient as described above is applied. This is the case in 2003 and 2008.

ii) If there was no snow cover present during the field survey, but before a later ALS campaign, the mass balance has been adjusted to the survey date by subtracting the amount of snow from the corresponding DTM DEM, as described in

 Sect. 4.1. This is the case for the years 2006 and 2007. The amount of snow determined agrees well for these years agrees well with extrapolated precipitation data using the altitudinal gradients between 5 rain gauges in the area from Vent.

iii) If snow was present during the field survey, but the ALS campaign had been conducted before the snowfall event, the mass of the snow cover measured during the field survey is added to the geodetic mass balance using the measured densities and the linear regression of snow probings for the elevation distribution. This is the case in 2002 and 2008.

iv) If snow was present during the field survey and the ALS data acquisition, the ALS-DTM ALS-DEM was adjusted regarding the snow cover conditions. When the ALS campaign was conducted after the field survey, the geodetic geodetically determined snow height is subtracted (section Sect. 4.1), and the mass of snow determined by field survey is added to the geodetic mass balance. This is the case for the years 2001, 2004, 2005, 2010.

There were no cases with snowfall between both surveys when the ALS data have been acquired before the field data. It is noted Note that two corrections have been applied for the year 2008 when the ALS data acquisition took place 21 days before the field survey and ablation as well as accumulation occurred . No survey date correction was necessary for in this period. For 2009 and 2011. 2011 no survey date corrections were necessary due to ALS-measurements very close to September 30$^{th}$.

**4.4 Representation of crevasses**

While crevasses are neglected in the glaciological method, they are partially resolved in the geodetic method. Although some crevasses might have been covered by snow during data acquisition, in all DTMs a number of big crevasses are visible , which open during the ablation season. However, depending in all DEMs. Depending on snow / melt conditions , crevasses are differently represented in the respective dDTMs, due to the ice movement between two ALS acquisitions and therefore have different impacts and their impact on ice movement, the recognition of crevasses in the single dDEMs and, hence, their impact on mass balance calculations . We varies widely. However, in this study we detected crevasses by assuming that they are deviations from a regular homogenous surface. By using the variance of elevation as a measure of terrain smoothness and by applying a closing filter, we derived a surface without crevasses (for detail we refer to Kodde et al. (2007) and Geist and Stötter (2010). Hence (Kodde et al., 2007; Geist and Stötter, 2010). Consequently, we calculated the volume change of a "crevasse free" glacier, to quantify differences possible uncertainties due to open crevasses $\epsilon_{crev}$ in the geodetic mass balance ($\epsilon_{crev}$; Table 5 Tab. 3).

**4.5 Internal and basal mass changes**

[revised manuscript text omitted]

The geodetic mass balance of Hintereisferner over the entire study period was mainly affected by snow being present in the year 2001 resulting in $\epsilon_{DTM}$ $\epsilon_{DEM}$ = +0.29 m w.e. Taking snow heights and densities in DTMs into account the effect of fresh snow on the DEMs of individual years into account (Sect. 4.1) leads to -0.41 $< \epsilon_{DTM} << \epsilon_{DEM} <$ +0.32 m w.e. (section 4.1). The value of

10    -0.41 m w.e. occurs in 2004/05 when snow was present at both ALS flight campaigns (Table 5 Tab. 3) making up for 37% of the initial mass balance value. Applying the workflow uncorrected mass change in this year.

Applying the work-flow for the spatially distributed density conversion (section Sect. 4.2) leads to $-0.04 < \epsilon_K < +0.31$ $-0.04 < \epsilon_{dc} < +0.31$ m w.e., with maxima in 2002/03 and 2005/06 (Table 5 Tab. 3). These maxima are due to the total lack of snow and firn at the end of these mass balance years. The uncertainty related to our density assumption (section 4.2) is between $\pm 0.01 < \sigma_K <$

15    $\pm 0.18$ Sect. 4.2) lies between $\pm 0.01 < \sigma_{dc} < \pm 0.18$ m w.e. with $\pm 0.22$ m w.e. over the entire period of record. As dates of the ALS campaigns diverge from the end of the hydrological year a survey date correction is required. Values for related adjustments

Values for adjustments related to survey date correction are in the order of $-0.08 < \epsilon_{survey} < +0.06$ $-0.08 < \epsilon_{sd} < +0.06$ m w.e. (section 4.3 and Table 5 Sect. 4.3 and Tab. 3). Significant melt amounts between ALS flight and field survey dates occur on small parts of the glacier tongue only. E.g. a nearly Ice ablation of almost 1 m ice ablation at the lowest stakes of Hintereisferner

20    measured between $30^{th}$ September (field survey) and $8^{th}$ October (ALS campaign) 2006 corresponds to a glacier wide specific mass loss of only 0.03 m w.e. during the same time. The differences Uncertainties related to the consideration of crevasses ($\epsilon_{crev}$) in the geodetic method are insignificantly small and vary between -0.04 and +0.06 m w.e. with +0.05 m w.e. for the 2001 to 2011 period (section Sect. 4.4 and Table 5). While the glacier wide effect of internal mass changes is small on an annual basis is($\epsilon_{int}$ = +0.05 m w.e. a$^{-1}$), it is significant on the decadal timescale (+0.50 m w.e.) (section 4.5 and Table 5 on the decadal timescale (Sect. 4.5 and

25    Tab. 3).

Annual totals for method-inherent differences ($\epsilon_{geod}$) are in the range of -0.40 to +0.57 m w.e. and accumulate to +0.28 m w.e. for the 2001 to 2011 period while the respective uncertainties are $\pm 0.07 < \sigma_{geod} < \pm 0.20$ and $\pm 0.51$ random uncertainties for individual years are $\pm 0.042 < \sigma_{geod} < \pm 0.183$ m w.e. for the cumulated values. The (Tab. 3. The geodetic balance calculated from the 2001 to and 2011 one step application of the geodetic method shows DEMs yields $\epsilon_{geod}$ = +0.77 m w.e. and $\sigma_{geod}$ = $\pm 0.20$ 0.34 m w.e. All applied corrections accounting

30    for method inherent differences numbers for the applied corrections and the single uncertainty sources ($\epsilon$ ) as well as numbers for related uncertainties (and $\sigma$) are summarized in Table 5. Figure 4 shows the vertical 3 while the altitudinal profiles of the now corrected glaciological and geodetic mass balances for each year from (2001/02 to 2010/11. The geodetic mass balance of Hintereisferner corrected for $\epsilon_{geod}$ for the ten years period 2001 to 2011 is -12.99 ±0.51 and -2.45 ±0.20 for the 2001 to 2011 one step analysis (Table 6) . In turn, the homogenized glaciological mass balance series (-12.04 ±0.65 )

is 0.95 and 0.31 less negative respectively. Figure 5 depicts the annual glaciological versus geodetic mass balances and their uncertainty ranges. All 11) are shown in Fig. 4.

**5.2 Methodological intercomparison**

The comparison of annual glaciological to geodetic balances shows that all but three annual data pairs match satisfyingly within the assessed uncertainty ranges (Fig. 2). The largest positive differences ($b_{glac} - b_{geod} = \Delta_b$ differences ($\Delta B = B_{glac.hom} - B_{geod.corr}$) between the two methods occur in the balance years 2002/03 with $\Delta_b$ $\Delta B$ = +0.92 m w.e. and 2005/06 with $\Delta_b$ $\Delta B$ = +0.60 m w.e. respectively. In 2006/07 the difference between glaciological and geodetic method is -0.45 m w.e., which means the geodetic result is less negative than the glaciological one. Note that the three years displaying the largest differences are at the same time the years with the most negative annual balances. Following Zemp et al. (2013) we perform a statistical significance test with The difference for the whole study period is 1.31 m w.e.. In order to detect significant biases between the two methods we calculated the reduced discrepancies ($\delta$) as described by Zemp et al. (2013) as

$$\delta = \frac{\Delta b}{\sqrt{\sigma_{glac}^2 + \sigma_{geod}^2}} \frac{\Delta B}{\sigma_{\text{common}}}, \tag{6}$$

where the term $\sqrt{\sigma_{glac}^2 + \sigma_{geod}^2}$ represents the common variance ($\sigma_{comvar}$) common variance $\sigma_{common}$ (Tab. 2) is defined as the RSS of the method-inherent uncertainties (Table 6 $\sqrt{\sigma_{glac}^2 + \sigma_{geod}^2}$). The more consistent the two methods, the closer $\delta$ is to zero and the null-hypothesis ($H_0$) on the 95% confidence level to ($H_{0_{95}}$) can be accepted. As $\delta$ falls within the 95% confidence interval ($\delta < 1.96$) for seven annual (all but 2002/03, 2005/06 and 2006/07) and the cumulative mass balance values, the two applied methods can be considered as statistically coherent. Hence, for these years, the glaciological method accurately captures the annual mass changes at Hintereisferner. similar (Tab. 2). Note that this approach is mainly designed for comparisons on longer (typical 10 years) time scales since biases on the annual scale might be missed. Nevertheless, in our case it allows the identification of significant deviations in three years.

From the common variance it is also possible to calculate the smallest bias that could theoretically be detected in the glaciological record (Zemp et al., 2013). The bias calculated at the 5% risk limit lies between 0.79 and 1.03 and far above the uncertainty of 0.76 and 0.96 m w.e. and is by far larger than the calculated uncertainty of annual glaciological balances of 0.21 in the glaciological balance measurements m w.e.. In contrast, the detectable bias decreases with the length of the analysed period, which can be explained by error propagation. However, it is not possible to statistically identify any biases that might explain the observed discrepancies in the mass balance years 2002/03, 2005/06 and 2006/07 (see Figure 5).

**6 Discussion**

In search for possible causes of these discrepancies large discrepancies between the methods in three of the sampled years, we explore the parameter space in which potential contribution of individual components of $\epsilon_{geod}$ vary. in the years of concern: The influence of temporary snow cover ($\epsilon_{DTM}$ $\epsilon_{DEM}$) on the geodetic mass balances is high and but a thorough consideration in our study ensures that the results are within the 95% confidence interval. In contrast, the survey date differences show little effect.

Concerning the conversion of glacier volume to mass changes, we used a new classification approach and a dimensionless conversion factor ($K$ to derive a more accurate value of annual conversion density ($\overline{\rho}$). Calculated values for $K$ correspond to densities $\overline{\rho}$ are in the range of 820-930 $\mathrm{kg\,m^{-3}}$. This is in line with a generally recommended the glacier-wide value of $850\pm60$ $\mathrm{kg\,m^{-3}}$ recommended by (Huss, 2013). Nevertheless, in 2010 $K$ $\overline{\rho}$ reaches 930 $\mathrm{kg\,m^{-3}}$, a value which at a first glance appears unrealistic. In this

5 year opposite signs of elevation changes in the accumulation and ablation area compensate for each other, which results in a conversion factor which is higher than the density of ice. Such is possible in cases of (i) short observation periods (1-3 years), (ii) small volume changes, (iii) strong year to year changes in the vertical mass balance gradients profiles, or combinations of these factors. Our approach accounts for year to year changes in the spatial extent and distribution of the snow/firn zones. Highest uncertainties arise in the years 2002/03 and 2005/06 when all snow from the previous winter melted entirely. As the

10 uncertainty associated with density is of particular importance (Moholdt et al., 2010; Huss, 2013) we conducted a sensitivity test for the periods of good agreement by holding all other parameters fixed. Densities calculated within our $K\overline{\rho}$-range (Table 5 Tab. 3) still lead to results within the 95% confidence interval.

As crevasses may influence geodetically calculated volume changes we assessed their impact on the geodetic method. The largest impact (0.06 $\mathrm{m\,w.e.}$, or 3% of glaciological mass balance) was detected for 2002/03 when numerous crevasses opened

15 due to the extremely hot summer causing extraordinary high glacier velocities (Geist et al., 2007). Hence, crevasses contribute negligibly to the differences between geodetic and glaciological mass balances.

Internal and basal fluxes processes are also of rather minor importance (-0.05 $\mathrm{m\,w.e.\,a^{-1}}$; section Sect. 4.5) and do not change the differences between the two data series substantially. Yet, we note that in years with extreme melt rates as in 2003 and 2006 meltwater penetrates the glacier body additional melt water from outside the glacier may enter the glacier bed in the tongue-area during

20 the ablation season and leads to the internal which leads to basal melt rates possibly exceeding the above estimate. . However However, even a doubling of our estimate to -0.10 $\mathrm{m\,w.e.\,a^{-1}}$ does not explain the large discrepancies between the glaciological and geodetic method in the years 2002/03, 2005/06 and 2006/07.

Other uncertainties possibly contributing to the high mass balance discrepancies in 2002/03, 2005/06 and 2006/07 may be method-inherent uncertainties related to the field measurements, such as the false determination of the last year's summer

25 surface. This might be an issue for the high discrepancies in the individual survey years, but cannot be quantified due to the lack of corresponding information. However, none of the discussed method-inherent uncertainties issues can explain the considerable high differences high deviations between glaciological and geodetic analyses in the mass balance years 2002/03, 2005/06 and 2006/07.

Nevertheless, a first hint for a potential reason is given by looking at the spatial mass balance distribution indicated by the altitudinal distribution

30 of point measurements as shown in Figure 6 Fig. 5 for the exemplary year 2002/03. In all three years of the poorly matched years, glaciological point data from elevations above 3000 m a.s.l. are basically missing on Hintereisferner , but all (Fig. 6). Given the glacier-median elevation of about 3039 m a.s.l. this means that the upper half of the glacier was not covered by measurements in these years. At the same time the three years of concern are among those with the most negative ones (Figure 7). mass balances within the Hintereisferner record (Fig. 2 and Tab. 2). The reason for missing measurements in higher

35 elevated areas in those years is the fact that no snow from the previous winter survived the warm summers at snow

pit locations and hence, traditional accumulation measurements were not possible. To address the problem of a mass balance network which had not been adapted in time, ablation rates measured at the highest stakes on the flat tongue (at about 3000 m a.s.l. and lower) had been multiplied with the observed ice exposure time of the higher slopes (G. Markl, personal communication). This disregards the impact of higher solar radiation intensity on the slopes compared to the flat

5 tongue, and the application of formerly observed "typical" spatial patterns of mass balance in the spatial extrapolations are considered to be possible reasons for the differences between the two methods in these years.

After several years of gradual degradation of the firn body, ice and older dark firn had suddenly become exposed over all altitude bands by mid of August 2003 with consequent effects on albedo and the surface energy budget. From then on, the The East and South facing high slopes of Hintereisferner had been exposed with a very low albedo exposed a low-albedo surface to high solar

10 radiation for 6 to 7 weeks of several weeks in the exceptionally warm and dry summer 2003 (Fink et al., 2004). As a consequence, the mass loss in the former accumulation area of Hintereisferner became large and almost constant above 2800 m a.s.l. unexpectedly large in areas without ablation stakes ($> 50\%$ of the glacier area). This effect had been observed on a smaller glacier some years earlier (Kaser et al., 2001). By facing this sudden change of the mass balance regime in 2002/03 and As a consequence, well known spatial patterns of surface melt of former years used in the mass balance network not being adapted in time, ablation rates measured at the highest stakes on the flat tongue (at about 3000 m a.s.l.) had been

15 multiplied with the observed ice exposure time of the higher slopes (G. Markl, personal communication). The thereby disregard of higher solar radiation intensity on the slopes compared to the flat tongue are considered to be a possible reason for the differences between the two methods analyses were no longer valid; an effect which had also been observed on a smaller glacier in the Eastern Alps some years earlier (Kaser et al., 2001).

While higher winter snow cover buried the dark ice surface far enough into the autumns of 2004 and 2005 the high glacier portions remained protected, even allowing obtaining protecting higher glacier portions and allowing for snow pits at the end of summer. In the hot July

20 of 2006 dark ice became again exposed and , the 2002/03 problem was repeated. became evident again in summer 2006 when dark glacier surfaces were again exposed after an early-summer heat wave.

In 2006/07 when the glaciological mass balance obtains more negative values than the geodetic one we face a different situation. In During summer 2007 there was a number of snow falls leading to high fall events increased the surface albedo in the upper part of Hintereisferner while stake measurements in the lower part of the glacier indicated relatively high ablation rates. The lack of metadata

25 We suspect that those high ablation rates were mistakenly extrapolated to higher elevations but the lack of meta-data for this particular year disables any further discussion and interpretation. In 2002/03, 2005/06 and 2006/07

However, based on our findings we argue for the geodetic data being closer to reality than the glaciological ones as recommended by Thibert et al. (2008) and Huss et al. (2009) in the years 2002/03, 2005/06 and 2006/07 (cf. Thibert et al., 2008; Huss et al., 2009). For all other years where when differences between the methods are statistically insignificant and where error bars overlap we keep

30 the glaciological data in the record. The crucial effect of replacing the three problematic years is well emphasized in the cumulative mass balance curves shown in Figure 8. the glaciological analyses yield plausible results. This interpretation is corroborated by comparison of the mass balance of Hinteresiferner with those of other glaciers in the region (see supplementary material).

Additional confidence for our approach comes from comparing the 2002/03, 2005/06 and 2006/07 mass balances of Hintereisferner with that of Silvrettagletscher (2.7 , Switzerland, 52 away), Jamtalferner (3.7 , Austria, 45 ), Weißbrunnferner (0.5 , Italy, 35 ) and Vernagtferner (7.9 , Austria, 6 ). While in the years 2002/03 and 2006/07 original Hintereisferner

35 values lay outside the spread of the other glaciers' mass balances and the reanalysed ones are inside, the 2005/06 originals are inside and the reanalysed value becomes the most

negative one in Figure 9. This is of no surprise with Hintereisferner being the lowest reaching glacier of all and among the most negative result in all analysed years. A more comprehensive discussion and justification for the different relative positions in Figure 9 would require a detailed investigation on local conditions including meteorological patterns for each individual glacier and mass balance year.

**7 Conclusions**

5 Over the past decades it has become a standard procedure to review the annual glaciological data alongside with decadal geodetic mass balances from a variety of sources Kuhn et al. (e.g., 1999); Hagg et al. (e.g., 2004); Cox and March (e.g., 2004); Thibert et al. (e.g., 2008); Huss et al. (e.g., 2009); Fischer (e.g., 2011); Galos et al. (e.g., 2017). None (e.g., Kuhn et al., 1999; Hagg et al., 2004; Cox and March, 2004; Thibert et al., 2008; Huss et al., 2009; Fischer, 2011; Galos et al., 2017). However, none of the mentioned studies uses annually obtained geodetic data series. (Geist et al., 2007) high-resolution ALS-data over one decade. Geist et al. (2007) were the first and so far only authors

10 comparing glaciological and ALS-based geodetic results on an annual timescale for time scale at Hintereisferner for the period 2001 to 2005. Their findings reveal considerable differences revealed considerable differences between the methods, especially in the year 2002/03. Yet, the study focuses on methodical issues only and does not aim at re-analysing the glaciologically obtained mass balances. It does neither include neither includes a thorough data homogenisation nor a robust uncertainty assessment and discussion.

In our review of the 2001 to 2011 Hintereisferner mass-balance record we showed that the consideration of method-inherent differences show that

15 the explicit consideration of uncertainty sources, such as the presence of snow cover, survey dates and density assumptions, is mandatory for accurately calculating annual geodetic mass balances. In turn Conversely, crevasses and internal processes seem not to play a key role. The largest potential source for differences between the geodetic and glaciological method on the annual scale is the presence of a snow cover . Our method allows us to correct during geodetic data acquisition. Although its reliance on a variety of raw data and meta information might limit its applicability to other sites or cases, our method allows correction for

20 method-inherent differences for every pixel and provides an appropriate basis for detecting discrepancies in the direct glaciological method. However, our reanalysis approach requires a variety of meta-information and raw data, which can limit its applicability to other sites or cases. However, the corrected Joint analysis of glaciological and geodetic data series show shows that the glaciological method in our case successfully captures the mass change in seven out of ten mass balance years and both methods generally agree on the annual as well as on the decadal time scale.

25 Our analysis further shows that in years with very negative mass balances and a low extent of the accumulation area, the glaciological measurement network has to be adapted accordingly. In the case of Hintereisferner, this means that additional ablation stakes in higher parts of the glacier are needed to properly assess the mass changes in regions where snow measurements could be performed in former times. Missing these changes , a resulting lack of respective data is often tried to overcome with different 
[revised manuscript text omitted]

Annual glaciological vs. geodetic mass balance. Both series are corrected for method-inherent differences and plotted with uncertainties (grey crosses). The black diagonal line marks equal balances from both methods.

[Figure]

**Figure 6.** Comparison of mass balances $(b_{glac}; b_{geod}(B_{glac.hom}$ and $B_{geod.corr})$ and their differences $(\triangle b \Delta B)$ with number of accumulation and ablation and accumulation measurements. Note that in areas higher than 3000 m a.s.l. only accumulation measurements were performed.

Calibration of glaciological mass balance series for the period 2001–2011 with the geodetic surveys for Hintereisferner. Cumulative adjusted mass balance $b_{glac}$ is calibrated with the geodetic mass change $(b_{geod})$ for the respective years 2002/03, 2005/06 and 2006/07 resulting in calibrated $b_{glac}$.

Comparison of original and reanalysed annual glaciological mass balances of Hintereisferner with different glaciers measured in the surrounding of Hintereisferner.

[revised manuscript text omitted]

---

## Author Response (AR2)

Dear Dr. Radić,

in the second revised version of our manuscript we carefully addressed all the points raised by Referee II. Equation 3 was corrected and the text was changed according to your remarks and those of the referee. Better rationale for this equation is now given in the text. We also corrected the text explaining Equation 5.

Please find our point-per-point response to the referee comments below. Again the comments by the referee are printed in *gray italics* and our response is given in blue regular font.

A marked-up version of the manuscript indicating all the changes applied to the paper is attached to our response letter.

Thank you for your consideration of our revised manuscript for publication in *The Cryosphere*.

Best regards,

Christoph Klug & Stephan P. Galos
(on behalf of the Co-authors)

**Author's response to referee-comments**

*You did a good job reorganizing the paper and addressing my concerns and those of Referee 1. Below, I've provided some minor editorial suggestions where needed. Do note that you need to revise text/equations (3) and (5), however.*

We revised Equation 3 and better explained and justified our approach of calculating the influence of random vertical DEM-uncertainty on the glacier wide geodetic mass balance.

We also revised the text explaining Equation 5.

*Abstract: Strike 'The validity of the results is critically assessed and concludes' and replace with 'We conclude that . . .'.*

Done.

*Pg. 1, line 21: Change sentence to, 'Long term records of surface mass balance often contain unquantifiable errors or biases that should be corrected prior to their use (Zemp et al., 2015).*

We revised the sentence, however, not fully following the referee's suggestion.

*Pg. 2, line 18: Change 'reanalize' to 'reanalyze' .*

Done.

*Pg. 2, line 18: Change 'This is achieved. . .' to 'We achieve this reanalysis through a. . .'*

Done.

*Pg. 3, line 2: Change 'amongst' to 'among'.*

Done.

*Pg. 7, line 7: Why is $\Delta V$ used to denote 'mass change' here. Volume times density is mass. I'm presuming you mean the latter? Revisions required here.*

We presume the referee refers to page 8, line 7 where we changed this error.

*Pg. 9, lines 1-2: Choose either 'C' or 'K' for reporting temperatures in manuscript.*

Done.

*Pg. 6, line 19: Write out all numbers less than 10 unless number is a measurement. (e.g. 1 cm but 'five stable control areas').*

Done.

*Pg. 6, lines25-30: Something is not correct with Equation 3. The summation is only for SD but not*

*for number of pixels? It is essential that the reader understand what you are doing here.*
We revised Equation 3 and better explained and justified our approach of calculating the influence of random vertical DEM-uncertainty on the glacier wide geodetic mass balance.

**Figures and Tables**

*Fig. 4 Caption: What does 'corrected' mean? Its inclusion in the caption is not required. Second sentence of caption isn't required (just ensure you make that point in the text and refer back to the figure).*
We omitted the word 'corrected' and the second sentence from the caption of this figure.

*Table 1. Provide units for flying height. Change 'Date of acquisition' to 'Acquisition date'*
Both points were changed.

**References**

[revised manuscript text omitted]